# AUTOENCODING-FREE CONTEXT COMPRESSION FOR LLMS VIA CONTEXTUAL SEMANTIC ANCHORS

**Xin Liu**[1,*]  **Runsong Zhao**[1,*]  **Pengcheng Huang**[1]  **Xinyu Liu**[1]  **Junyi Xiao**[1]
**Chunyang Xiao**  **Tong Xiao**[1,2,†]  **Shengxiang Gao**[3]  **Zhengtao Yu**[3]  **Jingbo Zhu**[1,2]

[1] School of Computer Science and Engineering, Northeastern University, Shenyang, China
[2] NiuTrans Research, Shenyang, China
[3] Kunming University of Science and Technology, Kunming, China

## ABSTRACT

Context compression is an advanced technique that accelerates large language model (LLM) inference by converting long inputs into compact representations. Existing methods primarily rely on autoencoding tasks to train special compression tokens to represent contextual semantics. While autoencoding tasks enable compression tokens to acquire compression capabilities, we remark that such capabilities potentially conflict with actual downstream task requirements, prevent the models from learning the features more beneficial for real-world usage. Based on this observation, we propose Semantic-Anchor Compression (SAC), a novel method that shifts from autoencoding task based compression to an architecture that is equipped with this compression capability *a priori*. Instead of training models to compress contexts through autoencoding tasks, SAC directly selects so-called anchor tokens from the original context and aggregates contextual information into their key-value (KV) representations. To ensure that anchors can effectively collect information, SAC introduces two key designs: (1) anchor embedding, a learnable embedding vector attached to the selected anchor tokens to mark compression carriers and (2) bidirectional attention modification, which enables anchor tokens to integrate information from the entire context. Experimental results show that SAC consistently outperforms existing context compression methods across different compression ratios and model sizes on question-answering and long-context summarization tasks. Our data, model and code have been released at https://github.com/lx-Meteors/SAC.

## 1 INTRODUCTION

The expanding scope of large language models (LLMs) to tasks such as processing long documents (Liu et al., 2024b; Li et al., 2024; Duan et al., 2025; Zhao et al., 2026), maintaining multi-turn dialogue coherence (Zhang et al., 2025; Yi et al., 2025; Guan et al., 2025), and generating responses grounded in extensive external knowledge (Lewis et al., 2020; Karpukhin et al., 2020; Huang et al., 2025) necessitates the incorporation of vast contexts into the model input. However, directly processing such extremely long contexts is fraught with challenges, including prohibitive computational costs, significant inference latency, and performance degradation, largely caused by the "lost-in-the-middle" phenomenon (Liu et al., 2024a).

To address these challenges, recent studies have proposed context compression (Chang et al., 2024; Li et al., 2025a; Lv et al., 2026; Tang et al., 2026a;b), a technique that typically appends special tokens (i.e. compression tokens) to the end of the context and leverages the LLM's causal attention mechanism to compress contextual information into a compact representation within these tokens. Once this compact representation is obtained, the LLM can generate responses conditioned on it, rather than being conditioned on the entire original context. The reduction in context length leads to substantial decreases in both inference time and GPU memory consumption. While effective, these approaches face a key limitation: the compression tokens are randomly initialized and lack inherent

---

[*]Equal contribution. Email: {liuxin1,zhaors}@mails.neu.edu.cn
[†]Corresponding author. Email: xiaotong@mail.neu.edu.cn

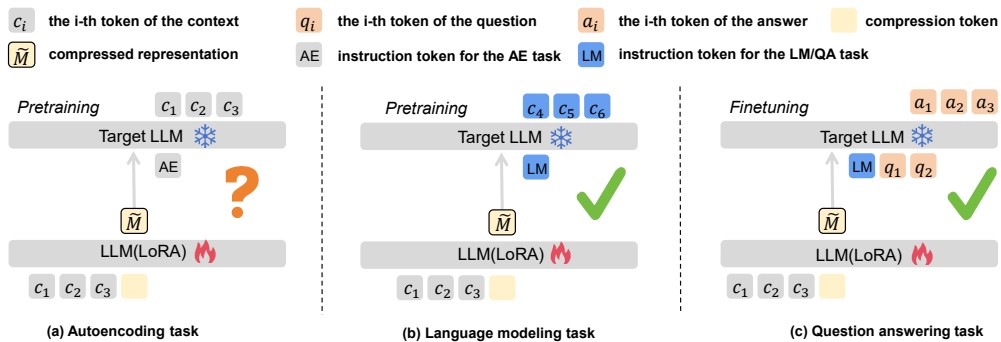

Figure 1: Three tasks for training the context compressor introduced by ICAE and followed by numerous works. The training uses (a) Autoencoding task and (b) Language modeling task to pretrain the encoder, then finetunes on (c) Question answering task.

semantic information. To address this limitation, prior works (Ge et al., 2024; Wang et al., 2024; Li et al., 2025b; Zhao et al., 2025; Tang et al., 2025b) typically rely on extensive pretraining on autoencoding (AE) and language modeling (LM) tasks (as shown in Fig. 1) to endow compression tokens with the ability to carry contextual information. However, while the model learns to reconstruct contexts from compression tokens through the AE task, such reconstruction objective can misalign with actual objectives (e.g. QA from compressd tokens) that downstream requires (we analyse such misalignment in subsection 5.1). Thus, this reliance on a suboptimal and costly pretraining stage raises a critical research question: is it possible to design a compression architecture that inherently understands context without a demanding AE phase?

To answer this question, our work introduces Semantic-Anchor Compression (SAC) (Figure 2), a novel architecture for the context compression task. Instead of appending new special tokens and requiring extensive autoencoding pretraining to learn their representations, SAC directly selects representative tokens from the original context to act as "anchor tokens" for compression. By leveraging these semantically meaningful anchors from the input itself, SAC incorporates natural semantic priors that obviate the need for autoencoding pretraining. To mark their special role, these selected tokens are augmented with dedicated "anchor embeddings", enabling the LLM to distinguish them from regular tokens. Furthermore, to enhance their compression capabilities, we modify the standard causal attention to a bidirectional attention mechanism. This allows anchor tokens to access information from the entire context, rather than being restricted to only preceding tokens. These modifications collectively foster a more effective context compression by providing anchor tokens with both distinct representations and comprehensive contextual awareness. To evaluate the effectiveness of SAC, we conduct experiments on question answering tasks and long-context summarization tasks, where it consistently outperforms strong baselines. Results also show that 1) our proposed method consistently improves over strong baselines across different compress ratios, 2) our proposed architecture achieves its best performance in a simpler training setting without autoencoding training, arguably because the anchor tokens already contain enough information about the original context. Our analysis reveals that SAC's compressed representations more closely resemble original context token KVs in feature space, which arguably enables LLMs to better understand them during inference.

## 2 RELATED WORKS

### 2.1 COMPRESSION METHOD

Many methods focus on reducing prompt lengths to improve LLM inference speed. CC (Wingate et al., 2022) utilizes contrastive learning to compress specific natural language prompts into shorter and unique soft prompt tokens. However, it cannot generalize to unseen prompts and requires retraining for new prompts. GIST (Mu et al., 2023) compresses original prompts into KV values through finetuning and can handle arbitrary unseen contexts. AutoCompressor (Chevalier et al., 2023) recursively combines compressed vectors with sub-prompts and aggregates all compressed vectors to

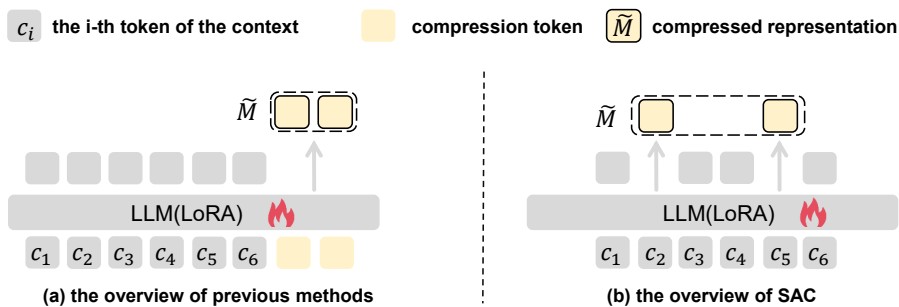

Figure 2: The difference between SAC and previous methods. While previous methods (a) compress contextual information into dedicated special tokens (referred to as compression tokens), SAC (b) compresses the context directly into the original contextual tokens themselves. Here, $\tilde{M}$ can represent either the output from the final layer of the LLM or the key-value pairs across all transformer layers, which are later used as compressed representations for LLM inference.

construct the final representation, enabling compression of longer contexts. However, both GIST and AutoCompressor require finetuning the LLMs (referred to later as target LLMs) to perform inference, which may affect LLMs' original capabilities.

ICAE (Ge et al., 2024) formulates context compression as training a general encoder that compresses contexts into compact representations understandable by target LLMs without finetuning target LLMs. To train the encoder, ICAE introduces autoencoding tasks and performs joint pretraining with language modeling tasks, followed by finetuning on downstream tasks. 500xCompressor (Li et al., 2025b) improves upon ICAE by replacing the compression carrier from the last layer output of compression tokens with KV values at each layer, achieving higher compression ratios. EPL (Zhao et al., 2025) identifies that ICAE and 500xCompressor neglect the impact of positional encoding and proposes distributing compression token position IDs uniformly across the entire context rather than placing them at the end. However, these methods still rely on autoencoding tasks to endow the compression tokens with the ability to carry contextual information.

Another category of prompt compression methods is based on token selection, which selects representative tokens from contexts based on token importance. SelectiveContext (Li et al., 2023), LLMLingua (Jiang et al., 2023), and LongLLMLingua (Jiang et al., 2024) employ causal small language models to evaluate token importance based on information entropy. LLMLingua-2 (Pan et al., 2024a) distills a token classifier to compute the probability of each token to be preserved. PerceptionCompressor (Tang et al., 2025a) preserves critical information at the token level while removing tokens that distract the LLM's attention. These works demonstrate that LLMs can understand original contexts using a small number of representative tokens. However, they do not perform compressed tokens training which limits the usability of the selected tokens by target LLMs. Our proposed SAC can be seen as a combination of token selection methods and compressed token training methods: it derives and train compressed representations that are based on tokens selected directly from the context and indeed is compatible with the token selection methods above.

## 2.2 BIDIRECTIONAL ATTENTION

Recent studies have shown that, removing the decoder's unidirectional causal constraint and introducing bidirectional attention can effectively enhance the model's representational capacity (Wang et al., 2020). For instance, NV-Embed (Lee et al., 2025) replaces causal attention with bidirectional attention during contrastive training, achieving strong performance on general text embedding and dense vector retrieval tasks. LLM2Vec (BehnamGhader et al., 2024), by enabling bidirectional attention alongside masked next-token prediction, significantly improves the model's ability to capture global semantics in text embedding tasks. These works indicate that bidirectional attention is advantageous for acquiring global semantic information and robust contextual representations. However, its effectiveness in context compression tasks remains underexplored. Motivated by these findings, we incorporate bidirectional attention into the compressor to enhance contextual modeling during the compression phase.

## 3 METHOD

### 3.1 TASK FORMULATION

Context compression is formally defined as follows: an encoder $\mathcal{E}$ compresses a context $C = (c_1, c_2, \ldots, c_{|C|})$ into a compact representation $\tilde{M}$ with $\tilde{M} = \mathcal{E}(C)$. Subsequently, a target LLM leverages the compressed representation $\tilde{M}$ in place of the original context $C$ to perform various tasks, such as question answering.

To train the encoder $\mathcal{E}$ to effectively extract contextual information, ICAE introduces three objective functions. The autoencoding loss $\mathcal{L}_{\mathbf{AE}}$ ensures that the compressed representation $\tilde{M}$ generated by $\mathcal{E}$ preserves all tokens in the context by learning to reconstruct the entire context $C$ regardless of the token relative importance, as shown in Figure 1a; mathematically, $\mathcal{L}_{\mathbf{AE}} = -\log P(C|\tilde{M})$. The language modeling loss $\mathcal{L}_{\mathbf{LM}}$ encourages $\tilde{M}$ to maintain predictive capability for future context $C' = (c_{|C|+1}, c_{|C|+2}, \ldots, c_{|C|+|C'|})$, enabling proactive information planning, as shown in Figure 1b; mathematically, $\mathcal{L}_{\mathbf{LM}} = -\log P(C'|\tilde{M})$. During pretraining, $\mathcal{L}_{\mathbf{AE}}$ and $\mathcal{L}_{\mathbf{LM}}$ are jointly optimized to obtain an initially effective encoder $\mathcal{E}$.

Additionally, during finetuning, the downstream task loss $\mathcal{L}_{\mathbf{QA}}$ (taking QA as an exemplar downstream task) enhances the ability of $\tilde{M}$ to extract information that is potentially relevant for downstream tasks. For example, for question answering (QA) tasks, the encoder learns through $\mathcal{L}_{\mathbf{QA}}$ to identify and preserve information in the context that is likely to be queried, enabling accurate answer generation $A = (a_1, a_2, \ldots, a_{|A|})$ when presented with subsequent questions $Q = (q_1, q_2, \ldots, q_{|Q|})$, as shown in Figure 1c; mathematically, $\mathcal{L}_{\mathbf{QA}} = -\log P(A|\tilde{M}, Q)$.

### 3.2 SEMANTIC-ANCHOR COMPRESSOR

A key distinction between our approach Semantic-Anchor Compression (SAC) and previous methods is that we derive compressed representations directly from selected context tokens, as shown in Figure 2. This involves selecting a subset of tokens $S$ from context $C$ as anchor tokens $S \subseteq C$. We believe that a good selection strategy benefits SAC. Following EPL, our default strategy divides the entire context $C$ into $|S|$ chunks and selects the middle token from each chunk. This setting helps maximize coverage of context $C$. As illustrated in Figure 3a, each selected token $c_i \in S$ is enhanced with the anchor embedding $e_A$, yielding an embedding sequence $E = (e_1, e_2, \ldots, e_{|C|})$:

$$e_i = \mathbf{Emb}(c_i) + \mathbf{1}_{c_i \in S} \cdot e_A \tag{1}$$

where $\mathbf{1}_{c_i \in S}$ is an indicator function that equals 1 when $c_i \in S$ and 0 otherwise. Following previous works, we employ a LLM with LoRA parameters $\theta_{LoRA}$ as the compressor: $\tilde{M} = \mathcal{E}(C) = \mathbf{LLM}(E|\theta_{LoRA})$. Overall, using original tokens from the context avoids learning compression tokens from scratch and we empirically validate that such design improves learning efficiency.

We notice that because the encoder uses causal attention, the anchor tokens $S$ do not have visibility to the full sentence, limiting their representation power. Hence in SAC, we modify the LLM to replace causal attention with bidirectional attention (see Figure 3b), to enhance the LLM's encoding capability. Notably, this bidirectional attention operates across all tokens, rather than being limited to the anchor tokens, allowing the model to capture richer contextual dependencies.

$\tilde{M}$ can be either the output of anchor tokens from the LLM's final layer or the key-value pairs from each layer. Following 500xCompressor, we use key-value pairs as the compressed representation $\tilde{M}$. For the encoder, we adopt the same language model as the decoder and leverage the original context's KV cache, enabling the decoder to comprehend the compressed representation more readily without requiring additional semantic alignment.[1] During pretraining, we only use $\mathcal{L}_{\mathbf{LM}}$ and do not use $\mathcal{L}_{\mathbf{AE}}$ to train the compressor. Following previous work, we use $\mathcal{L}_{\mathbf{QA}}$ for finetuning.

---

[1]For context compression, adopting different LLMs for encoder and decoder will create semantic gaps between the two and will impair performance, as noticed by (Lv et al., 2026).

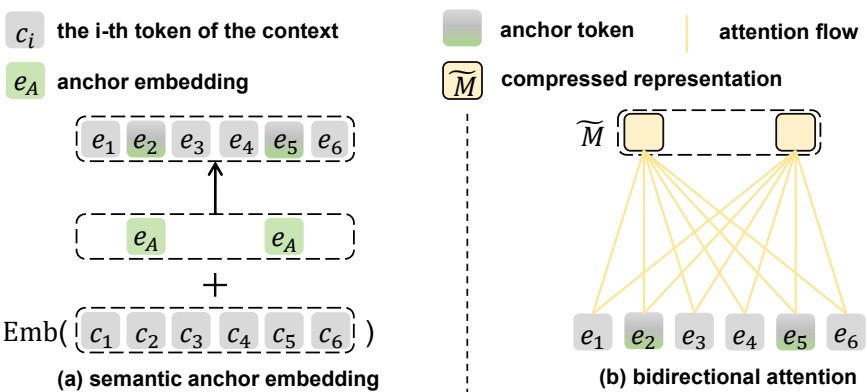

Figure 3: Key differentiators within SAC model architecture. (a) Representative tokens are transformed into anchor tokens through anchor embeddings. (b) The encoder in SAC adopts bidirectional attention, while the decoder operates with causal attention.

## 4 EXPERIMENTS

### 4.1 EXPERIMENTAL SETTING

**Dataset.** For continued pretraining, we use the corpus SlimPajama-6B (Soboleva et al., 2023). During finetuning and evaluation, we employ the standard MRQA (Fisch et al., 2019) question-answering dataset, which consolidates multiple QA tasks and standardizes them into a unified format. We evaluate SAC on both test sets, namely in-domain (ID) and out-of-domain (OOD), to comprehensively assess its in-distribution fitting ability and cross-domain generalization performance.

**Implementation Details.** SAC utilizes Llama-3.2-1B (Grattafiori & Dubey, 2024) as both the encoder and target LLM. We equip the encoder with trainable LoRA (Hu et al., 2022) adapters (rank = 128, $\alpha = 256$), while the target LLM parameters remain frozen. For each context, we partition it into sub-contexts of 510 tokens each. The compressor compresses each sub-context into a sub-compressed representation, and subsequently concatenates these sub-compressed representations to form the complete compressed representation. The number of anchor tokens $|S| = \lfloor L/r \rfloor$ is determined by the compression ratio $r$ and the length of the sub-context $L$. We train all models in two stages: pretraining for 20,000 optimization steps followed by finetuning for an additional 20,000 steps, both conducted with a batch size of 16. Complete hyperparameter configurations are provided in Appendix A.

**Baselines.** We use the Llama-3.2-1B model trained on the MRQA (Fisch et al., 2019) dataset as an uncompressed baseline (denoted as "Full-FT"). We compare our method against several context compression techniques: for hard compression, we choose LLMLingua-2 (Pan et al., 2024b) and evaluate its performance on the Full-FT model; for soft compression, we select ICAE (Ge et al., 2024), 500xCompressor (Li et al., 2025b), Activation Beacon (Zhang et al., 2024), DAST (Chen et al., 2025), and EPL (Zhao et al., 2025). To ensure a fair comparison, all these soft compression baselines are trained using the same experimental settings and data as our SAC method.

### 4.2 FINE-TUNING RESULTS

Tables 1 and 2 report the evaluation results of SAC on in-domain and out-of-domain MRQA datasets.

**Overall Performance.** For in domain results, SAC consistently outperforms all baseline models as shown in Table 1. At a 15× compression ratio, compared with the weaker ICAE and the stronger EPL, SAC shows a maximum improvement of 23.5% F1 / 26.8% EM and a minimum improvement of 6.7% F1 / 8.2% EM for in-domain evaluations. A similar trend can be observed for out-of-domain evaluations in Table 2 where the maximum improvement is 27.4% F1 / 28.9% EM, with a minimum improvement of 6.9% F1 / 9.2% EM compared to ICAE and EPL.

Table 1: For the fine-tuning results, we report in-domain performance on the MRQA datasets using ROUGE-1 F1 (Lin, 2004) and exact match (EM) (Maalouly, 2022).The compression ratio of Lingua-2 (Pan et al., 2024b) is set to 5×.

| Methods | SQuAD | | NewsQA | | TriviaQA | | SearchQA | | HotpotQA | | NQ | | Average | |
|---|---|---|---|---|---|---|---|---|---|---|---|---|---|---|
| | F1 | EM | F1 | EM | F1 | EM | F1 | EM | F1 | EM | F1 | EM | F1 | EM |
| Full-FT | 77.69 | 59.71 | 63.50 | 46.04 | 68.80 | 60.54 | 73.25 | 62.07 | 74.78 | 59.26 | 71.01 | 53.47 | 71.51 | 56.85 |
| Lingua-2 | 32.93 | 19.57 | 26.78 | 13.20 | 9.67 | 8.12 | 45.40 | 31.80 | 36.10 | 22.05 | 40.08 | 22.01 | 31.83 | 19.46 |
| *15x compression constraint* | | | | | | | | | | | | | | |
| ICAE | 31.90 | 18.91 | 25.25 | 11.97 | 51.78 | 42.94 | 64.81 | 52.89 | 45.22 | 30.32 | 48.01 | 30.67 | 44.50 | 31.28 |
| 500x | 40.68 | 24.97 | 32.01 | 16.76 | 53.84 | 44.86 | 65.65 | 53.70 | 53.01 | 36.30 | 50.93 | 33.26 | 49.35 | 34.98 |
| Beacon | 37.53 | 14.31 | 31.10 | 7.72 | 48.85 | 4.47 | 39.06 | 22.67 | 45.01 | 27.35 | 40.29 | 15.82 | 40.31 | 15.39 |
| DAST | 36.33 | 22.65 | 31.55 | 16.29 | 56.92 | 48.14 | 68.07 | 56.51 | 54.02 | 37.93 | 52.10 | 35.31 | 49.83 | 36.14 |
| EPL | 44.58 | 27.91 | 33.34 | 16.69 | 56.16 | 47.09 | 66.36 | 54.13 | 54.88 | 38.38 | 53.80 | 35.71 | 51.52 | 36.65 |
| **SAC** | **47.43** | **30.25** | **36.55** | **18.07** | **61.13** | **52.19** | **68.97** | **56.76** | **58.83** | **41.86** | **56.79** | **38.88** | **54.95** | **39.67** |

Table 2: For the fine-tuning results, we report out-of-domain performance on the MRQA datasets using ROUGE-1 F1 (Lin, 2004) and exact match (EM) (Maalouly, 2022).The compression ratio of Lingua-2 (Pan et al., 2024b) is set to 5×.

| Methods | BioASQ | | DROP | | DouRC | | RACE | | RE | | TQA | | Average | |
|---|---|---|---|---|---|---|---|---|---|---|---|---|---|---|
| | F1 | EM | F1 | EM | F1 | EM | F1 | EM | F1 | EM | F1 | EM | F1 | EM |
| Full-FT | 49.37 | 36.77 | 44.67 | 34.46 | 48.82 | 35.51 | 35.57 | 9.64 | 83.34 | 72.46 | 53.32 | 32.40 | 52.51 | 36.87 |
| Lingua-2 | 27.76 | 19.48 | 27.28 | 18.83 | 27.07 | 18.32 | 17.54 | 4.15 | 39.30 | 20.59 | 28.42 | 15.83 | 27.90 | 16.20 |
| *15x compression constraint* | | | | | | | | | | | | | | |
| ICAE | 35.51 | 24.47 | 30.39 | 21.96 | 13.78 | 9.06 | 15.21 | 3.71 | 55.24 | 40.33 | 34.75 | 21.56 | 30.81 | 20.18 |
| 500x | 36.30 | 25.93 | 33.46 | 23.55 | 20.53 | 12.72 | 18.49 | 3.41 | 54.37 | 41.11 | 41.09 | 25.82 | 34.04 | 22.09 |
| Beacon | 36.07 | 7.65 | 34.39 | 14.90 | **33.78** | 14.99 | **26.68** | 3.71 | 53.28 | 23.91 | 37.27 | 12.97 | 36.91 | 13.02 |
| DAST | 36.57 | 27.06 | 31.90 | 22.42 | 21.56 | 14.26 | 16.31 | 3.26 | 48.54 | 35.92 | 36.69 | 22.75 | 31.93 | 20.95 |
| EPL | 40.52 | 28.52 | 32.16 | 22.29 | 25.70 | 16.39 | 20.97 | 4.01 | 59.75 | 46.34 | 41.31 | 25.42 | 36.74 | 23.83 |
| **SAC** | **41.31** | **28.66** | **36.72** | **27.48** | 28.94 | **18.99** | 23.35 | **4.90** | **61.04** | **47.90** | **44.21** | **28.21** | **39.26** | **26.02** |

**Comparison with EPL.** As shown in Tables 1 and 2, EPL performs strongly overall thanks to its position layout designed for soft compression. Interestingly, SAC can be seen as building upon EPL [2] with enhanced compressed token representations introduced in Section 3. The overall results show that the token representation enhancement is effective as SAC outperforms EPL not only overall but also for each individual datasets in MRQA.

## 4.3 ABLATION STUDIES

In this subsection, we first show ablation studies on two SAC key components, namely anchor embeddings and bidirectional attentions. Then we study the effect of using different strategies to choose anchor tokens. Finally, we show empirically that our carefully designed SAC performs better without autoencoding training, a property that has motivated the SAC design.

Table 3: Component ablation results. We report the average F1/EM performance of the model on in-domain (ID) and out-of-domain (OOD) tasks after removing the bidirectional attention (w/o mask) and the anchor embedding (w/o anchor). Full results on all tasks are provided in the Appendix B.3, including Table 16 and Table 17.

| Methods | In-domain | | | | | | Out-of-domain | | | | | |
|---|---|---|---|---|---|---|---|---|---|---|---|---|
| | TriviaQA | | HotpotQA | | Average | | BioASQ | | TextbookQA | | Average | |
| | F1 | EM | F1 | EM | F1 | EM | F1 | EM | F1 | EM | F1 | EM |
| **SAC** | **65.06** | **55.93** | **67.41** | **50.28** | **66.24** | **53.11** | **44.66** | 31.45 | **52.24** | **32.93** | **48.45** | **32.19** |
| SAC(w/o mask) | 62.60 | 53.27 | 64.63 | 47.43 | 63.62 | 50.35 | 41.93 | 30.65 | 48.29 | 29.67 | 45.11 | 30.16 |
| SAC(w/o anchor) | 63.90 | 54.81 | 65.25 | 48.31 | 64.58 | 51.56 | 43.70 | **31.78** | 51.59 | 32.20 | 47.65 | 31.99 |

---

[2]This is because the anchor tokens selected by SAC share the same position IDs as the compressed tokens in EPL.

**Component Ablation.** As shown in Table 3, our ablation study demonstrates the critical roles of the bidirectional attention and anchor embedding. Removing either component results in significant performance degradation in both in-domain (ID) and out-of-domain (OOD) settings. The bidirectional attention mechanism enables anchor tokens to more effectively integrate information from the entire context, producing compressed representations that are more beneficial for downstream tasks. Meanwhile, the anchor embedding provides explicit structural signals that guide the model to accurately identify and process these anchor tokens, thereby ensuring the effectiveness of information compression.

**Token Selection.** Table 4 shows the results using different token selection strategies for SAC. The results indicate that random selection (*Random*) significantly degrades performance arguably for two reasons: the selected tokens lack semantic importance, and their random positions result in poor global coverage of the context, which together hinder effective representation. In contrast, information-based selection (*Lingua-2*) and our default strategy achieve on par results, and both substantially outperform existing baselines in Tables 1 and 2. This demonstrates that the SAC architecture can effectively leverage and enhance any high-quality token selection strategy, rather than relying on a specific choice, highlighting the generality and robustness of the SAC framework.

Table 4: Token selection results. Different token selection strategies are compared, including Random selection, Lingua-2-based selection (Pan et al., 2024b), and our uniform selection (Zhao et al., 2025). Average F1/EM scores are reported across in-domain (ID) and out-of-domain (OOD) tasks.

| Methods | ID-Average | | OOD-Average | |
|---|---|---|---|---|
| | F1 | EM | F1 | EM |
| **SAC** | **63.63** | **46.95** | **47.72** | **32.30** |
| SAC(Random) | 59.54 | 43.58 | 45.35 | 30.44 |
| SAC(Lingua-2) | 63.26 | 46.54 | 47.43 | 32.13 |

Table 5: Ablation study on the effects of autoencoding (AE) and language modeling (LM) objectives.

| Methods | In-domain Average | | | | | | Out-of-domain Average | | | | | |
|---|---|---|---|---|---|---|---|---|---|---|---|---|
| | 5x | | 15x | | 51x | | 5x | | 15x | | 51x | |
| | F1 | EM | F1 | EM | F1 | EM | F1 | EM | F1 | EM | F1 | EM |
| **SAC** | **63.63** | **46.95** | **54.95** | **39.67** | **46.37** | **33.08** | **47.72** | **32.30** | **39.26** | **26.02** | **32.24** | **21.44** |
| 500x(w/ LM only) | 53.23 | 38.70 | 49.76 | 35.71 | 44.46 | 31.41 | 38.22 | 25.73 | 33.99 | 22.05 | 30.09 | 18.99 |
| 500x(w/ AE+LM) | 55.26 | 40.14 | 49.35 | 34.98 | 43.19 | 30.26 | 38.46 | 25.40 | 34.04 | 22.09 | 30.43 | 20.09 |
| SAC(w/ AE only) | 56.55 | 40.34 | 49.93 | 35.33 | 43.95 | 30.64 | 42.08 | 27.98 | 35.50 | 23.29 | 28.77 | 18.32 |
| SAC(w/ AE+LM) | 62.04 | 45.80 | 51.73 | 36.67 | 44.69 | 31.37 | 47.26 | 32.25 | 37.23 | 23.96 | 31.01 | 19.90 |

**AE Effect.** Soft compression methods often use AE so that the compressed representations learn to reconstruct the original context. However, AE objective itself can be a limiting factor as the reconstruction target can potentially be misaligned with downstream tasks. SAC is designed with compressed representations directly derived from the contexts to eliminate the need for AE and we show our empirical investigations in Table 5. The experimental results show that training with only the AE objective leads to a substantial performance drop, and even when combined with the LM objective, the performance remains inferior to that of the full SAC model. It is worth noting that while ICAE (Ge et al., 2024) empirically shows that combining AE with LM tasks yields better results, the results do not generalize to other architectures such as 500xCompressor where Table 5 shows a mixed picture for performance. Overall, the results in Table 5 raise questions regarding the necessity of autoencoding tasks and suggest that autoencoding may not be entirely essential for context compression methods.

## 4.4 SCALABILITY RESULTS

To verify the generalizability of SAC across model scales, we conduct additional experiments on the larger Llama-3.2-3B and Llama-3.1-8B models. We show EPL as our baseline which performs the second best in main experiments shown in Table 1 and 2. As shown in Table 6, SAC outperforms EPL across all model sizes and evaluation settings, consistently achieving higher F1 and EM scores. On the 3B model, SAC achieves an average F1 score of 50.48 and an EM score of 34.73, outperforming the EPL baseline (47.46 / 31.82). On the 8B model, SAC scores 52.31 (F1) and 35.93 (EM), again

Table 6: We compare the performance of SAC and EPL across different model sizes ( Llama-3.2-3B and Llama-3.1-8B ) under a fixed 15× compression ratio. Evaluations are conducted in both in-domain (ID) and out-of-domain (OOD) settings, and we report ROUGE-1 F1 and Exact Match (EM) scores.

| Methods | In-domain | | | | | | Out-of-domain | | | | | |
|---|---|---|---|---|---|---|---|---|---|---|---|---|
| | SearchQA | | NQ | | Average | | DROP | | RE | | Average | |
| | F1 | EM | F1 | EM | F1 | EM | F1 | EM | F1 | EM | F1 | EM |
| EPL(3B) | 73.61 | 61.56 | 63.59 | 44.43 | 68.60 | 53.00 | 46.20 | 36.46 | 65.73 | 53.70 | 55.97 | 45.08 |
| **SAC(3B)** | **73.88** | **62.29** | **65.98** | **47.18** | **69.93** | **54.74** | **48.46** | **38.39** | **75.11** | **62.89** | **61.76** | **50.64** |
| EPL(8B) | 74.86 | 63.23 | 67.24 | 48.71 | 71.05 | 55.97 | 49.98 | 39.92 | 69.77 | 57.84 | 59.88 | 48.88 |
| **SAC(8B)** | **76.92** | **65.29** | **67.77** | **49.21** | **72.35** | **57.25** | **51.55** | **40.65** | **78.91** | **67.71** | **65.23** | **54.18** |

surpassing the EPL baseline (50.82 / 34.42). We remark that as the model goes larger, the gain does not seem to diminish. These consistent gains demonstrate that SAC effectively generalizes to larger model sizes and maintains its performance advantage. More detailed results on MRQA can be found in Tables 18 and 19.

## 4.5 SENSITIVITY ANALYSIS FOR COMPRESSION RATIOS

We conduct a detailed evaluation of model performance under different compression ratios (5×, 15×, and 51×), using the same experimental setup as in Table 1. The corresponding results are presented in Tables 7.

As expected, the F1 and EM scores of all methods decrease as the compression ratio increases from 5× to 51×, since higher compression ratios result in more information being discarded. At the highest compression ratio of 51×, the performance of different compression methods varies across datasets (see more detailed results in Appendix B.2, Tables 11 and 15): some methods perform well on certain datasets but underperform on others. Nonetheless, SAC consistently achieves the best average performance across all compression ratios, demonstrating strong robustness and stability.

Table 7: In-domain and out-of-domain performance under different compression ratios.

| Methods | In-domain Average | | Out-of-domain Average | |
|---|---|---|---|---|
| | F1 | EM | F1 | EM |
| *5x compression* | | | | |
| ICAE | 47.53 | 33.82 | 31.22 | 20.51 |
| 500x | 55.26 | 40.14 | 38.46 | 25.40 |
| EPL | 62.90 | 46.33 | 46.95 | 31.30 |
| **SAC** | **63.63** | **46.95** | **47.72** | **32.30** |
| *15x compression* | | | | |
| ICAE | 44.50 | 31.28 | 30.81 | 20.18 |
| 500x | 49.35 | 34.98 | 34.04 | 22.09 |
| EPL | 51.52 | 36.65 | 36.74 | 23.83 |
| **SAC** | **54.95** | **39.67** | **39.26** | **26.02** |
| *51x compression* | | | | |
| ICAE | 40.39 | 27.99 | 27.98 | 17.85 |
| 500x | 43.19 | 30.26 | 30.43 | 20.09 |
| EPL | 43.22 | 30.26 | 30.22 | 19.48 |
| **SAC** | **46.37** | **33.08** | **32.24** | **21.44** |

## 4.6 LONG-CONTEXT SUMMARIZATION RESULTS

To further validate the robustness and usefulness of SAC, we conduct additional experiments on two long-context summarization tasks using the QMSum (Zhong et al., 2021) and GovReport (Huang et al., 2021) datasets. In these experiments, the models are configured with a maximum input length of 32K tokens and are trained using

Table 8: We compare the performance of SAC and EPL on long-context summarization tasks, including QMSum (Zhong et al., 2021) and GovReport (Huang et al., 2021), reporting ROUGE-1 F1 scores.

| Methods | QMSum | GovReport | Average |
|---|---|---|---|
| EPL | 14.82 | 20.40 | 17.61 |
| **SAC** | **14.95** | **22.03** | **18.49** |

datasets with a $15\times$ compression ratio. Both SAC and EPL models are trained and evaluated on the corresponding datasets.

The results in Table 8 show that SAC maintains strong performance on long-context summarization tasks, surpassing EPL across all datasets. Because EPL is trained with AE objective, the results suggest that removing the AE objective for SAC does not impair the model's capability to model long contexts. Moreover, SAC is able to capture sufficient global information to support tasks that require comprehensive understanding of the entire context, highlighting its adaptability and generalization performance in long-context generation scenarios. More details are provided in Appendix B.5.

## 5 ANALYSIS

### 5.1 AE ANALYSIS

We attribute the negative impact of the AE task on SAC primarily to the following two related factors.

**Intuitive Perspective.** The AE objective requires the compressed representations to reconstruct the complete context, forcing the model to encode a large number of tokens within its limited capacity; some tokens might require significant model memories while not useful for downstream tasks, thereby weakening model's ability to retain task-critical information.

**AE-LM Gradient Perspective.** In Figure 9, we visualize the gradient cosine similarity between the autoencoding and the language modeling losses during training. The figure shows that while the two gradients largely align at the very beginning, the cosine similarity quickly approaches zero, suggesting that the two tasks are largely orthogonal in parameter space. This orthogonality causes the optimization of autoencoding to interfere with language modeling updates, thereby hindering downstream task performance. [3]

### 5.2 ATTENTION VISUALIZATION

To understand the unique behavior of compressed models, we analyze the attention patterns of the final layer at a $5\times$ compression rate.

At a lower 5× compression rate, as shown in Figure 6, SAC attention map presents a clear positive diagonal, indicating that its compressed tokens primarily attend to local tokens, adhering to the attention prior similar to EPL. In contrast, the attention map of 500xCompressor appears more diffused which arguably complicates learning and impairs its performance. As the compression rate increases (as shown in Figure 7 and 8), EPL attention pattern starts also to diffuse while SAC exhibits a focused attention pattern, with its anchor tokens attending to only a few key original context tokens. Hypothetically, the focused attention pattern retains important token information to handle downstream tasks and explains the superior performance we empirically observe.

### 5.3 REPRESENTATION ANALYSIS

**Key Representation Analysis.** In the key representation space (see Figure 4 up), the compression tokens (orange) of SAC and EPL are distributed relatively close to the context tokens (blue), while the compression tokens of 500xCompressor are clearly separated from the context tokens. This discrepancy arises from the architectural design of each method. EPL modifies the positional IDs of its additional compression tokens to share the same rotational angle (RoPE) as the original context tokens, thereby reducing the distance between them; without such modifications, 500xCompressor compression tokens do not align with context token key representations. In SAC, the anchor tokens are directly embedded within the original context, in consequence their representations maintain close semantic ties with the context tokens.

**Value Representation Analysis.** In the value representation space (see Figure 4 bottom), the anchor tokens of SAC are uniformly distributed across all regions with the value representations of the context tokens, without forming independent sub-clusters. This suggests that SAC's anchor embedding strategy allows for compressed value representations that more closely match the distribution of the

---

[3]The autoencoding performance will also be impacted by the LM loss for the same orthogonality reason. We show empirically the AE performance in Appendix Table 14.

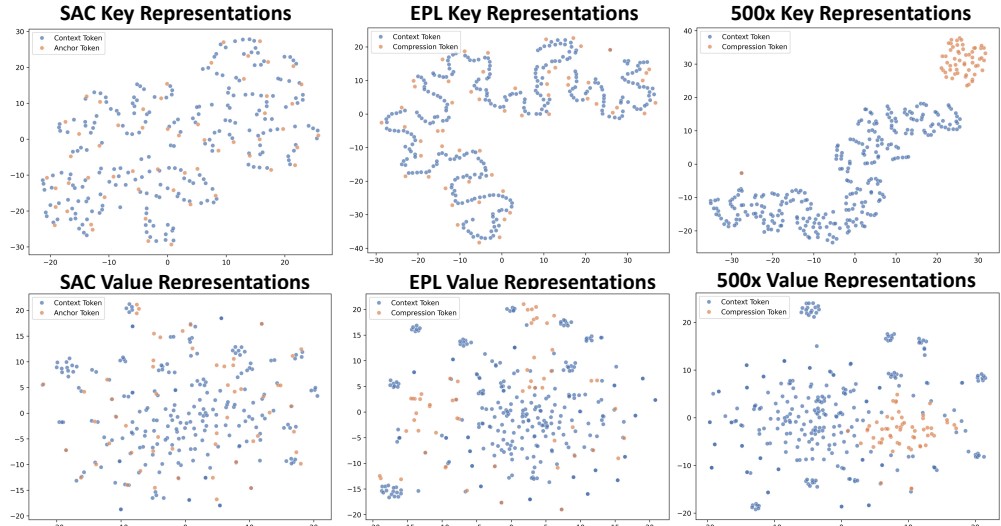

Figure 4: The t-SNE visualization shows the key representations of the final layer KV values for SAC, 500xCompressor (Li et al., 2025b), and EPL (Zhao et al., 2025), respectively.

original value space. In contrast, although EPL's compression tokens also overlap with the context tokens, they overlap less than SAC's: the compression tokens appear relatively sparse in the core regions and show a slight clustering tendency at the boundaries. This indicates that EPL's value representations still exhibit a degree of semantic shift relative to the original value space, which is even more pronounced in the 500xCompressor.

## 6 CONCLUSION

In this paper, we propose a novel, autoencoding-free context compression method, **Semantic-Anchor Compression (SAC)**. Unlike traditional context compression approaches, SAC does not rely on training compression tokens to reconstruct the original input. Instead, it directly selects representative *anchor* tokens from the context and aggregates contextual information into their key-value (KV) representations via a bidirectional attention mechanism. This approach effectively compresses lengthy contexts while avoiding any impairment to the language model's original language modeling capabilities, while traditional context compression can hinder performance by introducing additional compression tokens and the autoencoding tasks, as our analysis shows. Experimental results show that SAC achieves high compression rates while significantly outperforming existing methods across multiple question answering and long-context summarization tasks, demonstrating strong cross-task generalization and an effective balance between compression efficiency and model performance.

## 7 REPRODUCIBILITY STATEMENT

We declare that the work presented in this paper is reproducible. Our data, model, and code have been released at https://github.com/lx-Meteors/SAC. This code can be used to reproduce the experimental results. The repository includes detailed instructions for environment setup, running experiments, data processing, and result evaluation.

## 8 ACKNOWLEDGEMENTS

This work was supported in part by the National Science Foundation of China (Nos. 62276056 and U24A20334), the Yunnan Fundamental Research Projects (No.202401BC070021), the Yunnan Science and Technology Major Project (No. 202502AD080014), the Fundamental Research Funds for the Central Universities (Nos. N25BSS054 and N25BSS094), and the Program of Introducing Talents of Discipline to Universities, Plan 111 (No.B16009).

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

# A   EXPERIMENT DETAILS

We perform pretraining and fine-tuning using bf16 precision on 8 NVIDIA RTX 3090 GPUs (24GB). For pretraining, we randomly sample data from the SlimPajama-6B dataset with a token length ranging from 510 to 2040. This data is then split into two halves: one for the auto-encoding (AE) task and the other for the language modeling (LM) task (the AE half is discarded for models without the AE objective). For downstream tasks, we process the MRQA dataset into a (Context, Question, Answer) format for finetuning. Detailed hyperparameters can be found in Table 9.

Table 9: Hyperparameters for training

| Hyperparameter | Value |
| --- | --- |
| Optimizer | AdamW |
| Betas | (0.9, 0.95) |
| Weight decay | 0.1 |
| Learning rate | 1e-4 (pretrain) |
|  | 5e-5 (finetuning) |
| Scheduler | Constant |
| Batch size | 16 |
| Warmup | 300 |
| Training steps | 20k (pretrain) |
|  | 20k (finetuning) |
| Clip norm | 2.0 |

# B   DETAILED RESULTS

## B.1   PRETRAINING RESULTS

As shown in Table 10, our method, SAC, achieves the lowest perplexity (10.79) among all baseline models. This suggests that removing the autoencoding (AE) objective in SAC allows the model to better focus on the language modeling task, thereby improving its predictive capability. Furthermore, since SAC avoids the additional computational overhead from independent compression tokens and the AE task, its training is approximately 31% faster than ICAE and 26% faster than 500xCompressor and EPL.

Table 10: Pretraining comparison of SAC and existing context compression methods, results on LM perplexity and training time.

| Methods | LM-PPL | Training Time(hours) |
| --- | --- | --- |
| ICAE | 12.35 | 3.85 |
| 500x | 11.83 | 3.60 |
| EPL | 10.88 | 3.60 |
| **SAC** | **10.79** | **2.66** |

## B.2   FINE-TUNING RESULTS

Tables 11 and 15 report the evaluation results of SAC on in-domain and out-of-domain MRQA datasets, which we analyze from three perspectives: overall performance, effect of compression ratio, and domain generalization.

**Overall Performance.** SAC consistently outperforms all baselines across a variety of conditions, including compression ratios, and both in-domain and out-of-domain tests, as shown in Tables 11 and 15. Averaging the results of the context compression methods across different compression ratios, SAC shows a maximum improvement of 24.6% F1 / 28.6% EM and a minimum improvement of 4.6% F1 / 5.7% EM in in-domain evaluations. For out-of-domain tests, the maximum improvement is 32.5% F1 / 36.2% EM, with a minimum improvement of 4.6% F1 / 6.9% EM.

**Impact of Compression Ratio.** We conduct a detailed evaluation of model performance under different compression ratios (5×, 15×, and 51×), as shown in Tables 11 and 15. As expected, F1 and EM scores of all methods decrease with increasing compression ratio, from 5× to 51×, since higher compression ratios result in more information being discarded. At the highest compression rate of 51×, the performance of different compression methods is not consistent. While one method may perform well on certain datasets, it may underperform on others. Nonetheless, SAC consistently achieves the best average performance.

**Cross-Domain Generalization.** We evaluate the generalization capability of SAC on out-of-domain datasets, as shown in Table 15. Under all compression ratio constraints, SAC consistently achieves the highest average F1/EM scores among all methods. Specifically, at a 5× compression ratio, SAC attains average F1 and EM scores of 47.72 and 32.30, outperforming the second-best EPL method by 0.77 and 1.0 points, respectively. At a more challenging 15× compression ratio, SAC achieves average F1 and EM scores of 39.26 and 26.02, surpassing EPL by 2.52 and 2.19 points, with an EM improvement approaching 10%. Even at an extreme 51× compression ratio, SAC maintains average F1 and EM scores of 32.24 and 21.44, still leading EPL by 2.02 and 1.96 points, respectively. These results indicate that the compressed representations learned by SAC exhibit strong cross-domain robustness.

Table 11: For the finetuning results, we report in-domain performance using ROUGE-1 F1 (Lin, 2004) and exact match (EM) (Maalouly, 2022) scores on the following datasets: SQuAD (Rajpurkar et al., 2016), NewsQA (Trischler et al., 2017), TriviaQA (Joshi et al., 2017), SearchQA (Dunn et al., 2017), HotpotQA (Yang et al., 2018), and NaturalQuestions (NQ) (Kwiatkowski et al., 2019).

| Methods | SQuAD | | NewsQA | | TriviaQA | | SearchQA | | HotpotQA | | NQ | | Average | |
|---|---|---|---|---|---|---|---|---|---|---|---|---|---|---|
| | F1 | EM | F1 | EM | F1 | EM | F1 | EM | F1 | EM | F1 | EM | F1 | EM |
| Full-FT | 77.69 | 59.71 | 63.50 | 46.04 | 68.80 | 60.54 | 73.25 | 62.07 | 74.78 | 59.26 | 71.01 | 53.47 | 71.51 | 56.85 |
| Lingua-2 | 32.93 | 19.57 | 26.78 | 13.20 | 9.67 | 8.12 | 45.40 | 31.80 | 36.10 | 22.05 | 40.08 | 22.01 | 31.83 | 19.46 |
| *5x compression constraint* | | | | | | | | | | | | | | |
| ICAE | 36.20 | 22.12 | 28.06 | 13.77 | 54.63 | 45.59 | 65.12 | 53.06 | 48.79 | 33.40 | 52.36 | 34.99 | 47.53 | 33.82 |
| 500x | 51.62 | 33.63 | 39.70 | 22.63 | 57.62 | 48.76 | 66.43 | 54.38 | 59.10 | 42.20 | 57.11 | 39.26 | 55.26 | 40.14 |
| EPL | 64.72 | 44.28 | 48.74 | **27.45** | 63.75 | 54.54 | 69.69 | 57.73 | 67.16 | 49.79 | 63.32 | 44.16 | 62.90 | 46.33 |
| **SAC** | **65.37** | **44.83** | **49.39** | 27.14 | **65.06** | **55.93** | **69.99** | **58.06** | **67.41** | **50.28** | **64.56** | **45.44** | **63.63** | **46.95** |
| *15x compression constraint* | | | | | | | | | | | | | | |
| ICAE | 31.90 | 18.91 | 25.25 | 11.97 | 51.78 | 42.94 | 64.81 | 52.89 | 45.22 | 30.32 | 48.01 | 30.67 | 44.50 | 31.28 |
| 500x | 40.68 | 24.97 | 32.01 | 16.76 | 53.84 | 44.86 | 65.65 | 53.70 | 53.01 | 36.30 | 50.93 | 33.26 | 49.35 | 34.98 |
| EPL | 44.58 | 27.91 | 33.34 | 16.69 | 56.16 | 47.09 | 66.36 | 54.13 | 54.88 | 38.38 | 53.80 | 35.71 | 51.52 | 36.65 |
| **SAC** | **47.43** | **30.25** | **36.55** | **18.07** | **61.13** | **52.19** | **68.97** | **56.76** | **58.83** | **41.86** | **56.79** | **38.88** | **54.95** | **39.67** |
| *51x compression constraint* | | | | | | | | | | | | | | |
| ICAE | 26.17 | 14.58 | 22.48 | 9.69 | 47.62 | 39.23 | 64.31 | 52.80 | 38.91 | 24.78 | 42.87 | 26.86 | 40.39 | 27.99 |
| 500x | 30.09 | 17.11 | 25.06 | 12.20 | 50.84 | 42.13 | 64.92 | 53.29 | 42.15 | 27.32 | 46.07 | 29.53 | 43.19 | 30.26 |
| EPL | 30.09 | 17.49 | 24.49 | 11.54 | 51.15 | 42.38 | 65.12 | 53.16 | 42.19 | 27.23 | 46.29 | 29.77 | 43.22 | 30.26 |
| **SAC** | **31.81** | **18.78** | **27.36** | **13.56** | **56.73** | **47.85** | **65.82** | **53.76** | **48.28** | **32.84** | **48.22** | **31.70** | **46.37** | **33.08** |

## B.3 ABLATION RESULTS

In the main text, we discuss the significant performance gains of SAC over all baseline methods. To provide more detailed evidence, we present the full ablation study results here. As shown in Table 16 and Table 17, our conclusion holds not only in terms of average performance but is also consistently validated on each individual dataset.

## B.4 SCALABILITY RESULTS

We use the same data and training settings as described in the paper, with a compression ratio of 15. The training is conducted on 8 A100 GPUs for Llama-3.2-3B and Llama-3.1-8B.

The results are shown in Tables 18 and 19. Compared with the stronger baseline EPL, SAC achieved the best average performance in all cases, demonstrating its ability to effectively scale up to larger model sizes while maintaining its performance advantage.

### B.5 LONG-CONTEXT SUMMARIZATION RESULTS

To further investigate whether this degradation phenomenon is specific to QA or reflects a broader conflict between AE and downstream tasks, we additionally evaluate SAC without AE and SAC with AE on other long-context tasks. The evaluation metric used is ROUGE-F1.

For the summarization task, we train SAC and EPL on QMSum (Zhong et al., 2021) and GovReport (Huang et al., 2021) with a maximum input length of 32K tokens. The test results are shown in Table 20. The results indicate that SAC achieves the highest average performance on long-document summarization tasks. Notably, the training data size of QMSum (1.26k) is much smaller than that of GovReport (17.5k), which may explain why SAC(ae+lm) performs slightly better on QMSum, while SAC performs better on the larger GovReport dataset. We hypothesize that as redundant information increases in long contexts and the information capacity of compressed tokens is limited, the AE objective of reconstructing all information may impose additional burden on the model, thereby negatively affecting performance.

We further evaluate on a long-context question answering benchmark with a context length of 24K tokens, using the model checkpoints from Table 1, which are trained with a 15× compression ratio on 2K-token contexts. The results are shown in Table 21. On both single-document and multi-document question answering tasks, SAC consistently outperforms the best baseline compression methods under a 15× compression ratio. These results indicate that our method can maintain its performance advantages even when applied to longer contexts.

### B.6 FLOPS RESULTS

We want to clarify a critical advantage of SAC: while SAC does enable bidirectional attention for anchor tokens, it does not require appending additional $k$ special tokens to the sequence like 500xCompressor. This means SAC operates on shorter sequences during inference, directly reducing computational overhead.

Let's quantify this advantage. Given a context with shape $[b, s, h]$ where $b$ is batch size, $s$ is sequence length, $h$ is hidden size, and $I$ is the FFN intermediate size, we compare the theoretical FLOPs:

Table 12: Comparison of FLOPs between SAC and 500xCompressor for different modules.

| Modules | SAC-FLOPs | 500x-FLOPs |
|---|---|---|
| $\mathbf{x}(\mathbf{W_Q}/\mathbf{W_K}/\mathbf{W_V})$ | $3 \cdot 2bsh^2$ | $3 \cdot 2b(s+k)h^2$ |
| $\mathbf{QK^T}$ | $2bs^2h$ | $b(s+k)^2h$ |
| $\mathbf{AV}$ | $2bs^2h$ | $b(s+k)^2h$ |
| $\mathbf{xW_O}$ | $2bsh^2$ | $2b(s+k)h^2$ |
| $\mathbf{X_{out}W_{up}}$ | $2bshI$ | $2b(s+k)hI$ |
| $\mathbf{X_{out}W_{gate}}$ | $2bshI$ | $2b(s+k)hI$ |
| $\mathbf{X_{out}W_{down}}$ | $2bshI$ | $2b(s+k)hI$ |
| $\mathbf{sum}$ | $bhs(8h+4s+6I)$ | $bh(s+k)[8h+2(s+k)+6I]$ |

Concrete comparison: With typical settings ($b$=1, $s$=510, $h$=2048, $I$=8192):

- At 5× compression($k$=102): 500xCompressor requires **1.19× more FLOPs** than SAC
- At 10× compression($k$=51): 500xCompressor requires **1.08× more FLOPs** than SAC

To validate the theoretical analysis, we additionally measured empirical inference latency. Using a single GPU with batch size 10 over 1000 sequences, we recorded the compression time for both methods. The results are reported in Table 13.

Table 13: Empirical compression time comparison (ms per batch) between SAC and 500xCompressor.

| Methods | Compression Time |
|---|---|
| 500x | 257.43 |
| **SAC** | **243.87** |

These results show that SAC not only reduces theoretical FLOPs but also achieves lower empirical compression latency. Despite the use of bidirectional attention, SAC is more efficient than 500xCompressor, as the computational cost of bidirectional attention is offset by operating on the original, shorter sequence.

## C  VISUALIZATION ANALYSIS

### C.1  TRAINING CURVES ANALYSIS

Figure 5 shows the training loss curves at different compression ratios on the MRQA dataset. The training loss of our SAC model consistently converges better than other baseline methods across all compression ratios, which demonstrates that the compressed representations obtained from the SAC architecture are more beneficial for language modeling tasks. Notably, as the compression ratio increases appropriately, the difference in convergence between SAC and the other baselines becomes more significant.

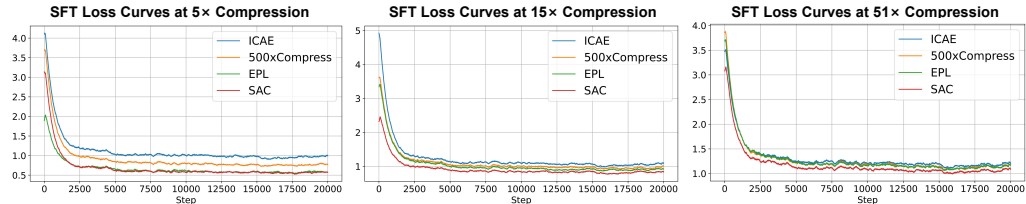

Figure 5: Supervised finetuning loss curves. The figure illustrates the training loss trajectories of different models under three compression ratios: 5×, 15×, and 51×.

## D  THE USE OF LARGE LANGUAGE MODELS

We used a large language model (LLM) as a general-purpose assist tool. The LLM's primary role was in assisting with writing and text editing, such as refining prose and correcting grammar and spelling to ensure the paper's professionalism and fluency. We explicitly state that the LLM was not involved in the core ideation or methodological design of this research. All core contributions of the paper, including the proposal of the methodology, the construction and execution of experiments, and the analysis of results, were performed independently by the authors.

Table 14: Analysis of autoencoding (AE) reconstruction and language modeling performance under different training objectives.

| Methods | AE-PPL | LM-PPL | AE-BLEU4 |
|---|---|---|---|
| ICAE | 4.08 | 12.35 | 12.02 |
| 500xCompress | 1.09 | 11.83 | 87.21 |
| EPL | 1.03 | 10.88 | 97.50 |
| SAC(ae+lm) | 1.03 | 11.01 | 98.19 |
| SAC(only ae) | 1.00 | - | **99.94** |
| **SAC** | - | **10.79** | - |

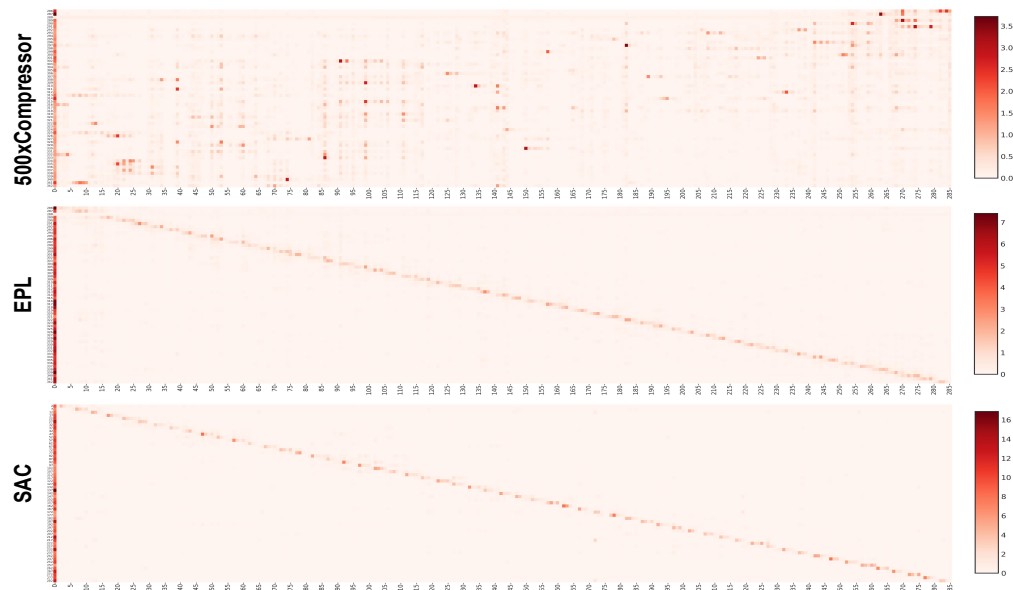

Figure 6: Attention maps of different models finetuned under a 5× compression rate. From top to bottom, the figure displays the final layer attention maps for the 500xCompressor, EPL, and SAC models, respectively. The x-axis represents the original context tokens, and the y-axis represents the compression tokens.

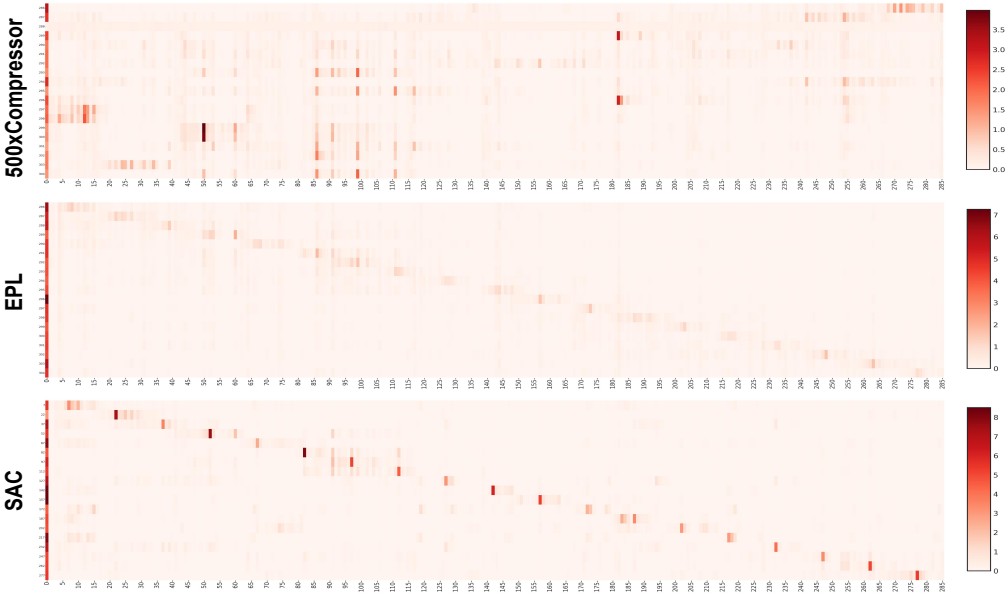

Figure 7: Attention maps of different models finetuned under a 15× compression rate. From top to bottom, the figure displays the final layer attention maps for the 500xCompressor, EPL, and SAC models, respectively. The x-axis represents the original context tokens, and the y-axis represents the compression/anchor tokens.

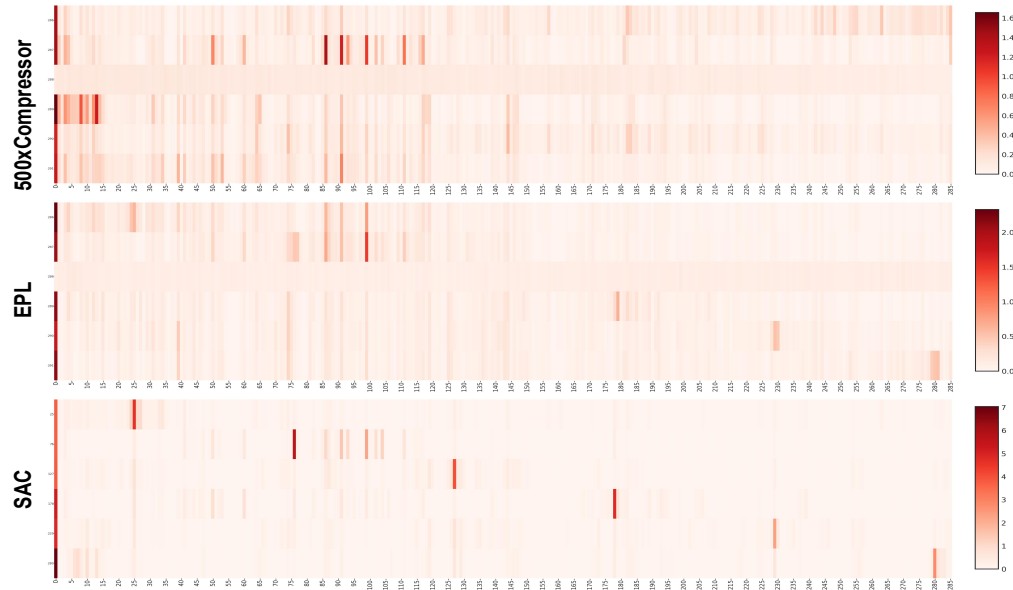

Figure 8: Attention maps of different models finetuned under a 51× compression rate. From top to bottom, the figure displays the final layer attention maps for the 500xCompressor, EPL, and SAC models, respectively. The x-axis represents the original context tokens, and the y-axis represents the compression tokens.

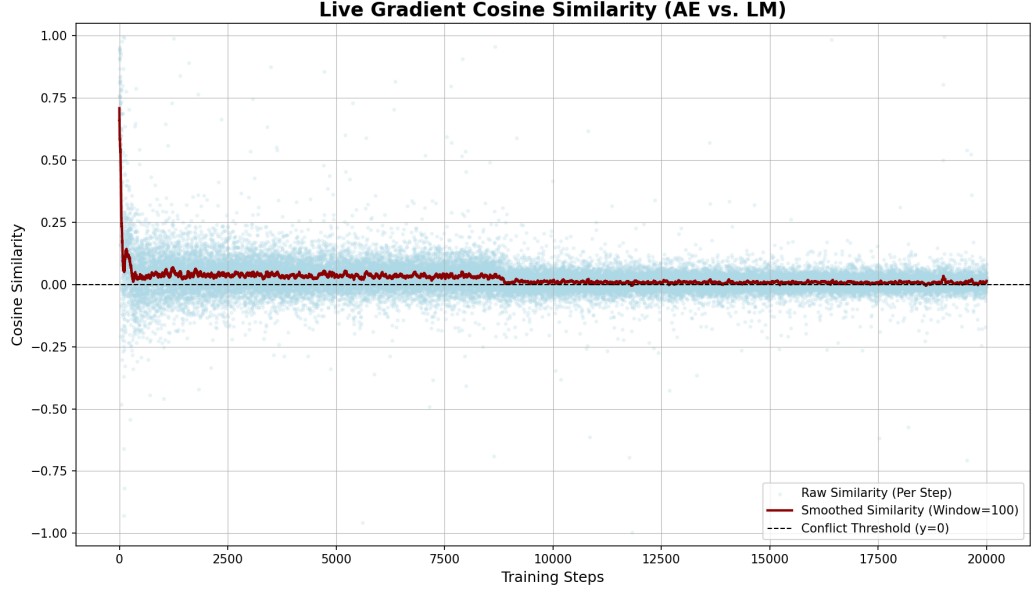

Figure 9: Gradient Cosine Similarity between AutoEncoding (AE) Loss and Language Modeling (LM) Loss.

Table 15: For the finetuning results, we report out-of-domain performance using ROUGE-1 F1 and exact match (EM) scores on the following datasets: BioASQ (Tsatsaronis et al., 2015), DROP (Dua et al., 2019), DuoRC (Saha et al., 2018), RACE (Lai et al., 2017), Relation Extraction (RE) (Levy et al., 2017), and TextbookQA (TQA) (Kembhavi et al., 2017).

| Methods | BioASQ | | DROP | | DouRC | | RACE | | RE | | TQA | | Average | |
|---|---|---|---|---|---|---|---|---|---|---|---|---|---|---|
| | F1 | EM | F1 | EM | F1 | EM | F1 | EM | F1 | EM | F1 | EM | F1 | EM |
| Full-FT | 49.37 | 36.77 | 44.67 | 34.46 | 48.82 | 35.51 | 35.57 | 9.64 | 83.34 | 72.46 | 53.32 | 32.4 | 52.51 | 36.87 |
| Lingua-2 | 27.76 | 19.48 | 27.28 | 18.83 | 27.07 | 18.32 | 17.54 | 4.15 | 39.30 | 20.59 | 28.42 | 15.83 | 27.90 | 16.20 |
| *5x compression constraint* | | | | | | | | | | | | | | |
| ICAE | 36.08 | 26.06 | 28.95 | 21.09 | 16.67 | 10.79 | 15.65 | 3.12 | 54.73 | 41.01 | 35.24 | 20.96 | 31.22 | 20.51 |
| 500x | 40.30 | 28.99 | 35.40 | 25.55 | 29.43 | 19.32 | 21.57 | 4.90 | 65.43 | 50.88 | 38.62 | 22.75 | 38.46 | 25.40 |
| EPL | **46.05** | **32.58** | 39.94 | 28.94 | 39.10 | **27.12** | **30.99** | 6.08 | 76.07 | 62.31 | 49.54 | 30.74 | 46.95 | 31.30 |
| **SAC** | 44.66 | 31.45 | **41.55** | **30.87** | **39.48** | 26.92 | 30.53 | **6.23** | **77.87** | **65.40** | **52.24** | **32.93** | **47.72** | **32.30** |
| *15x compression constraint* | | | | | | | | | | | | | | |
| ICAE | 35.51 | 24.47 | 30.39 | 21.96 | 13.78 | 9.06 | 15.21 | 3.71 | 55.24 | 40.33 | 34.75 | 21.56 | 30.81 | 20.18 |
| 500x | 36.30 | 25.93 | 33.46 | 23.55 | 20.53 | 12.72 | 18.49 | 3.41 | 54.37 | 41.11 | 41.09 | 25.82 | 34.04 | 22.09 |
| EPL | 40.52 | 28.52 | 32.16 | 22.29 | 25.70 | 16.39 | 20.97 | 4.01 | 59.75 | 46.34 | 41.31 | 25.42 | 36.74 | 23.83 |
| **SAC** | **41.31** | **28.66** | **36.72** | **27.48** | **28.94** | **18.99** | **23.35** | **4.90** | **61.04** | **47.90** | **44.21** | **28.21** | **39.26** | **26.02** |
| *51x compression constraint* | | | | | | | | | | | | | | |
| ICAE | 33.82 | 23.67 | 27.94 | 19.29 | 11.14 | 6.86 | 14.89 | 3.41 | 47.02 | 34.02 | 33.08 | 19.83 | 27.98 | 17.85 |
| 500x | 32.17 | 23.07 | **30.11** | **21.76** | 13.42 | 8.53 | 15.18 | 2.67 | **54.62** | **41.86** | 37.10 | 22.62 | 30.43 | 20.09 |
| EPL | 32.52 | 22.21 | 29.64 | 20.89 | 13.16 | 8.13 | **17.15** | 3.12 | 53.72 | 40.37 | 35.15 | 22.16 | 30.22 | 19.48 |
| **SAC** | **36.95** | **26.86** | 29.52 | 20.89 | **21.85** | **14.26** | 15.87 | **4.00** | 48.19 | 36.43 | **41.05** | **26.21** | **32.24** | **21.44** |

Table 16: We conduct ablation studies for SAC under a 5× compression rate on the in-domain dataset in three sets: component ablation, token selection, and the influence of the autoencoding (AE) task.

| Methods | SQuAD | | NewsQA | | TriviaQA | | SearchQA | | HotpotQA | | NQ | | Average | |
|---|---|---|---|---|---|---|---|---|---|---|---|---|---|---|
| | F1 | EM | F1 | EM | F1 | EM | F1 | EM | F1 | EM | F1 | EM | F1 | EM |
| *Component Ablation* | | | | | | | | | | | | | | |
| SAC | 65.37 | 44.83 | 49.39 | 27.14 | 65.06 | 55.93 | 69.99 | 58.06 | 67.41 | 50.28 | 64.56 | 45.44 | 63.63 | 46.95 |
| SAC(w/o mask) | 60.21 | 39.93 | 45.93 | 25.74 | 62.60 | 53.27 | 66.66 | 54.82 | 64.63 | 47.43 | 61.53 | 42.55 | 60.26 | 43.96 |
| SAC(w/o anchor) | 61.69 | 41.72 | 46.52 | 25.45 | 63.90 | 54.81 | 68.03 | 56.17 | 65.25 | 48.31 | 62.21 | 43.88 | 61.27 | 45.06 |
| *Token Selection* | | | | | | | | | | | | | | |
| SAC(Random) | 58.16 | 39.00 | 42.94 | 22.98 | 62.37 | 53.17 | 69.13 | 57.36 | 63.86 | 46.70 | 60.76 | 42.26 | 59.54 | 43.58 |
| SAC(Lingua-2) | 64.89 | 44.28 | 48.92 | 27.11 | 64.55 | 55.13 | 69.89 | 58.04 | 67.05 | 49.74 | 64.23 | 44.93 | 63.26 | 46.54 |
| *AE Effect* | | | | | | | | | | | | | | |
| 500x(w/ LM only) | 44.71 | 28.89 | 37.24 | 20.39 | 58.97 | 50.19 | 65.67 | 53.74 | 56.74 | 40.52 | 56.07 | 38.45 | 53.23 | 38.70 |
| 500x(w/ AE+LM) | 51.62 | 33.63 | 39.70 | 22.63 | 57.62 | 48.76 | 66.43 | 54.38 | 59.10 | 42.20 | 57.11 | 39.26 | 55.26 | 40.14 |
| SAC(w/ AE only) | 56.98 | 37.60 | 41.09 | 20.61 | 58.19 | 49.08 | 64.02 | 51.65 | 61.58 | 44.13 | 57.23 | 38.98 | 56.55 | 40.34 |
| SAC(w/ AE+LM) | 64.68 | 44.62 | 46.64 | 25.62 | 63.34 | 54.27 | 68.40 | 56.48 | 66.61 | 49.72 | 62.56 | 44.06 | 62.04 | 45.80 |

Table 17: We conduct ablation studies for SAC under a 5× compression rate on the out-of-domain dataset in three sets: component ablation, token selection, and the influence of the autoencoding (AE) task.

| Methods | BioASQ | | DROP | | DouRC | | RACE | | RE | | TQA | | Average | |
|---|---|---|---|---|---|---|---|---|---|---|---|---|---|---|
| | F1 | EM | F1 | EM | F1 | EM | F1 | EM | F1 | EM | F1 | EM | F1 | EM |
| *Component Ablation* | | | | | | | | | | | | | | |
| SAC | 44.66 | 31.45 | 41.55 | 30.87 | 39.48 | 26.92 | 30.53 | 6.23 | 77.87 | 65.40 | 52.24 | 32.93 | 47.72 | 32.30 |
| SAC(w/o mask) | 41.93 | 30.65 | 40.24 | 28.48 | 36.48 | 23.58 | 28.21 | 5.49 | 69.09 | 55.63 | 48.29 | 29.67 | 44.04 | 28.92 |
| SAC(w/o anchor) | 43.70 | 31.78 | 40.55 | 30.34 | 36.97 | 25.58 | 30.05 | 6.82 | 75.88 | 62.35 | 51.59 | 32.20 | 46.46 | 31.51 |
| *Token Selection* | | | | | | | | | | | | | | |
| SAC(Random) | 45.94 | 34.31 | 40.87 | 30.21 | 35.35 | 24.12 | 29.28 | 5.64 | 70.11 | 56.34 | 50.53 | 32.00 | 45.35 | 30.44 |
| SAC(Lingua-2) | 44.49 | 31.91 | 41.50 | 29.61 | 39.47 | 26.58 | 29.96 | 7.12 | 77.67 | 65.47 | 51.46 | 32.07 | 47.43 | 32.13 |
| *AE Effect* | | | | | | | | | | | | | | |
| 500x(w/ LM only) | 43.54 | 33.11 | 35.40 | 25.82 | 27.71 | 17.59 | 19.73 | 3.86 | 62.31 | 48.27 | 40.60 | 25.75 | 38.22 | 25.73 |
| 500x(w/ AE+LM) | 40.30 | 28.99 | 35.40 | 25.55 | 29.43 | 19.32 | 21.57 | 4.90 | 65.43 | 50.88 | 38.62 | 22.75 | 38.46 | 25.40 |
| SAC(w/ AE only) | 40.85 | 29.39 | 35.32 | 25.28 | 31.55 | 21.32 | 25.86 | 4.90 | 72.29 | 57.90 | 46.61 | 29.08 | 42.08 | 27.98 |
| SAC(w/ AE+LM) | 44.84 | 32.31 | 41.47 | 31.14 | 39.29 | 27.58 | 30.11 | 6.23 | 77.12 | 64.42 | 50.74 | 31.87 | 47.26 | 32.26 |

Table 18: Experimental Results of the 1B/3B/8B Models on the In-Domain QA Tasks.

| Methods | SQuAD | | NewsQA | | TriviaQA | | SearchQA | | HotpotQA | | NQ | | Average | |
|---|---|---|---|---|---|---|---|---|---|---|---|---|---|---|
| | F1 | EM | F1 | EM | F1 | EM | F1 | EM | F1 | EM | F1 | EM | F1 | EM |
| EPL(1B) | 44.58 | 27.91 | 33.34 | 16.69 | 56.16 | 47.09 | 66.36 | 54.13 | 54.88 | 38.38 | 53.80 | 35.71 | 51.52 | 36.65 |
| **SAC(1B)** | **47.43** | **30.25** | **36.55** | **18.07** | **61.13** | **52.19** | **68.97** | **56.76** | **58.83** | **41.86** | **56.79** | **38.88** | **54.95** | **39.67** |
| EPL(3B) | 62.43 | 42.65 | 46.72 | 26.14 | 68.83 | 59.92 | 73.61 | 61.56 | 68.50 | 51.03 | 63.59 | 44.43 | 63.95 | 47.62 |
| **SAC(3B)** | **63.11** | **43.51** | **49.23** | **28.80** | **69.65** | **60.85** | **73.88** | **62.29** | **68.87** | **51.40** | **65.98** | **47.18** | **65.12** | **49.01** |
| EPL(8B) | **65.79** | **45.55** | 51.18 | 28.96 | **72.40** | 63.49 | 74.86 | 63.23 | 70.98 | 53.65 | 67.24 | 48.71 | 67.08 | 50.60 |
| **SAC(8B)** | 64.92 | 44.80 | 49.77 | **29.20** | 72.37 | **63.70** | **76.92** | **65.29** | 70.80 | **54.25** | **67.77** | **49.21** | 67.09 | **51.08** |

Table 19: Experimental Results of the 1B/3B/8B Models on the Out-of-Domain QA Tasks.

| Methods | BioASQ | | DROP | | DouRC | | RACE | | RE | | TQA | | Average | |
|---|---|---|---|---|---|---|---|---|---|---|---|---|---|---|
| | F1 | EM | F1 | EM | F1 | EM | F1 | EM | F1 | EM | F1 | EM | F1 | EM |
| EPL(1B) | 40.52 | 28.52 | 32.16 | 22.29 | 25.70 | 16.39 | 20.97 | 4.01 | 59.75 | 46.34 | 41.31 | 25.42 | 36.74 | 23.83 |
| **SAC(1B)** | **41.31** | **28.66** | **36.72** | **27.48** | **28.94** | **18.99** | **23.35** | **4.90** | **61.04** | **47.90** | **44.21** | **28.21** | **39.26** | **26.02** |
| EPL(3B) | 46.76 | 32.98 | 46.20 | 36.46 | 36.81 | 25.32 | 33.80 | 7.57 | 65.73 | 53.70 | 55.46 | 34.86 | 47.46 | 31.82 |
| **SAC(3B)** | **47.96** | **34.51** | **48.46** | **38.39** | **42.21** | **29.85** | 33.61 | 7.72 | **75.11** | **62.89** | 55.53 | **35.00** | **50.48** | **34.73** |
| EPL(8B) | **52.52** | **37.57** | 49.98 | 39.92 | 40.09 | 27.24 | 35.22 | 7.27 | 69.77 | 57.84 | 57.36 | **36.66** | 50.82 | 34.42 |
| **SAC(8B)** | 48.54 | 34.04 | **51.55** | **40.65** | **41.44** | **29.18** | **35.52** | 7.72 | **78.91** | **67.71** | **57.87** | 36.26 | **52.31** | **35.93** |

Table 20: Model Performance on 32K Long-Context Summarization Tasks.

| Methods | QMSum | GovReport | Average |
|---|---|---|---|
| **SAC** | 14.95 | **22.03** | **18.49** |
| SAC(ae+lm) | **15.32** | 20.15 | 17.74 |
| EPL | 14.82 | 20.40 | 17.61 |

Table 21: Model Performance on 24K Long-Context QA Tasks.

| Methods | MultiFieldQA | NarrativeQA | Qasper | Sigle-Doc-Avg. | 2WikiMQA | MuSiQue | HotpotQA | Multi-Doc-Avg. |
|---|---|---|---|---|---|---|---|---|
| **SAC** | **17.75** | **11.33** | **11.67** | **13.58** | **24.78** | **12.36** | **26.03** | **21.06** |
| SAC(ae+lm) | 17.38 | 10.49 | 9.05 | 12.31 | 22.47 | 7.76 | 23.88 | 18.04 |
| EPL | 14.34 | 10.50 | 7.58 | 10.81 | 21.99 | 8.43 | 23.72 | 18.05 |

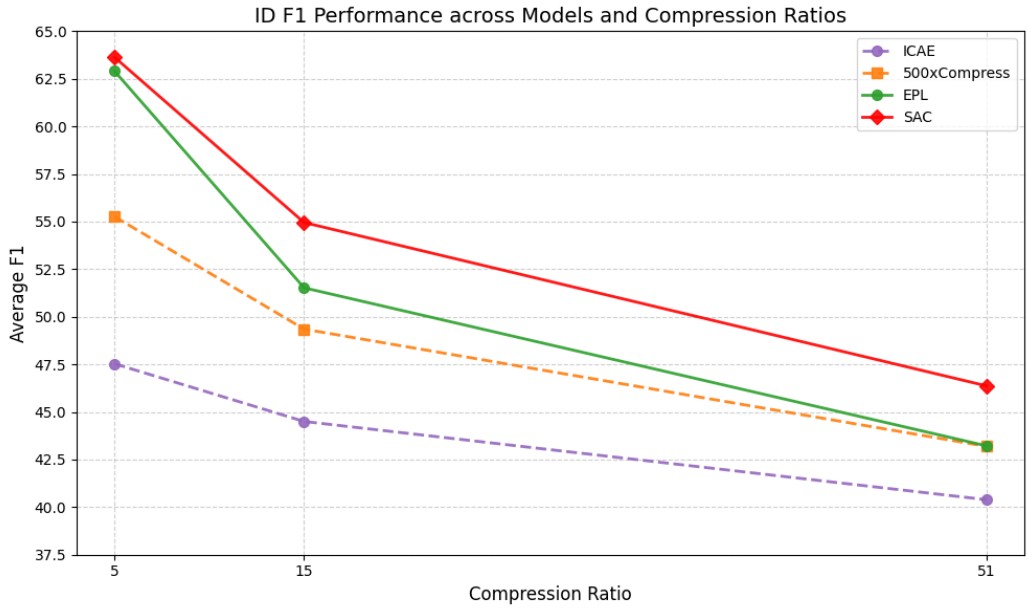

Figure 10: Efficiency and Performance Trade-off Curves on In-Domain (ID) Tasks.

