# OpenReview forum: "Autoencoding-Free Context Compression for LLMs via Contextual Semantic Anchors"
_ICLR.cc/2026/Conference — ICLR 2026 Poster_

### Official Review · Reviewer_4eQ5 · 2025-10-20

**Soundness:** 2
**Presentation:** 3
**Contribution:** 2
**Rating:** 2
**Confidence:** 5

**Summary:**

The paper proposes Semantic-Anchor Compression (SAC), which directly selects anchor tokens from the original context and aggregates contextual information into KV representations. It consists of two key designs---anchor embeddings and bidirectional attention to ensure the compression performance.

**Strengths:**

1. The proposed solution is simple but effective. The author shows good results on many downstream tasks.
2. The paper is generally well-written and easy to follow.

**Weaknesses:**

1. The intuition behind the proposed method is confusing. The authors argue that "SAC is distinct to previous compression-based methods" because "previous methods compress contextual information into context-agnostic special tokens". However, in a auto-regressive setting, the compressed token is actually context-dependent since it can see all tokens behind in attention computation.
2. SAC add lora to Llama-3.2-1B with attention map modification, from causal to bidirection attention. This sounds quite confusing and seems not to be a normal routine. Why does not use encoder models directly?
3. The idea of context compression has been well-studied in many previous works, such as [1]-[5]. These works impair the novelty of proposed method.

[1] Yen, Howard. Long-context language modeling with parallel context encoding. MS thesis. Princeton University, 2024.

[2] Li, Yuhong, et al. "Snapkv: Llm knows what you are looking for before generation." Advances in Neural Information Processing Systems 37 (2024): 22947-22970.

[3] Zhang, Zhenyu, et al. "H2o: Heavy-hitter oracle for efficient generative inference of large language models." Advances in Neural Information Processing Systems 36 (2023): 34661-34710.

[4] Han, Wei, et al. "Two are better than one: Context window extension with multi-grained self-injection." arXiv preprint arXiv:2410.19318 (2024).

[5] Zhang, Peitian, et al. "Long context compression with activation beacon." arXiv preprint arXiv:2401.03462 (2024).

**Questions:**

see weakness

---

> ### Author Response · Authors · 2025-11-21
> **Response by Authors to 4eQ5 (1/2)**
>
> Dear Reviewer 4eQ5,
>
> Thank you for your valuable time and constructive feedback. We appreciate your positive comments on the simplicity and effectiveness of our method, as well as the clarity of our writing. We will address your concerns below.
>
> # Response to Weaknesses
>
> ## For W1
>
> > The intuition behind the proposed method is confusing. The authors argue that "SAC is distinct to previous compression-based methods" because "previous methods compress contextual information into context-agnostic special tokens". However, in a auto-regressive setting, the compressed token is actually context-dependent since it can see all tokens behind in attention computation.
>
> We agree that the **compressed representations** produced by the encoder are context-dependent. However, in our paper, the terms **"compression tokens"** or **"special tokens"** refer specifically to the initial representations fed into the encoder, which are inherently context-agnostic before encoding (SAC, in the contrary selects token embeddings from the context and thus is no more context-agnostic). We apologize for the confusion caused by this terminology and will clarify this distinction in our revised version.
>
> ## For W2
>
> > SAC add lora to Llama-3.2-1B with attention map modification, from causal to bidirection attention. This sounds quite confusing and seems not to be a normal routine.
>
> Yes, we agree that modifying the causal attention of an LLM to bidirectional attention is unconventional, and this is indeed one of the main distinctions between our method and previous approaches [1,2,3], which all use LLama's original causal attention map in the encoders.  Our empirical results in Table 1 & Table 2 show that SAC outperforms previous approaches [1,2,3] and the ablation study (Table 3) demonstrates that this modification to bidirectional attention yields significant improvements.  Moreover, we observe also that recently there are some studies using LLMs and modifying their causal attention into bidirectional attention [4,5,6] to construct powerful text encoders;  our modification is inspired by their success.
>
> > Why does not use encoder models directly?
>
> The primary reason is to maintain comparability with prior work [1,2,3]. Additionally, since the compressed representations in SAC and [2,3] are KV pairs, the compressor must have the same number of layers as the target LLM. For ICAE [1], using encoder models is feasible. Below are the results using ALBERT [7] and Llama-3.2-1B as compressors (The detailed experimental results can be found in **Supplementary Results: Effect of Different Compressors**.):
>
> ||ID|OOD|Avg|
> |-|-|-|-|
> |SAC|**54.95**|**39.26**|**47.11**|
> |ICAE-Llama|44.50|30.81|37.66|
> |ICAE-ALBERT|15.81|15.84|15.83|
>
> As shown in the table, ICAE with ALBERT as the compressor significantly underperforms ICAE with Llama, which can be attributed to the substantially smaller model size and less extensive pretraining of ALBERT compared to Llama-3.2-1B.

---

> ### Author Response · Authors · 2025-11-21
> **Response by Authors to 4eQ5 (2/2)**
>
> ## For W3
>
> > The idea of context compression has been well-studied in many previous works, such as [8]-[12]. These works impair the novelty of proposed method.
>
> We appreciate the reviewer's comment regarding related work. Our contribution is novel within the specific paradigm of architecture-preserving context compression.
>
> The cited works represent different technical routes:
> - **[9,10]** are training-free and perform KV cache pruning rather than semantic compression.
> - **[8,11,12]** modify the decoder architecture by adding attention modules.
>
> In contrast, SAC operates as a pure preprocessing method that compresses context into standard LLM prompts without any modification to the decoder's architecture or parameters. This enables SAC to pre-compress frequently used documents for already deployed LLMs, providing a practical advantage over methods requiring decoder modification and fine-tuning [8,11,12].
>
> Compared to other preprocessing compressors [1,2,3], SAC introduces three key innovations:
> 1. Semantic-Aware Anchor Selection: Direct selection of representative tokens from the original context, enhanced with specialized anchor embeddings.
> 2. Bidirectional Context Aggregation: Effective integration of contextual information through a bidirectional attention mechanism.
> 3. End-Task Oriented Training: Model optimization focused on language modeling tasks and downstream tasks, eliminating the need for autoencoding objectives.
>
> These technical distinctions directly enable the performance improvements demonstrated in our experiments.
>
> We would be happy to compare with methods from different technical routes to provide more comprehensive insights. However, given the limited rebuttal timeline, we conducted a comparison with **Activation Beacon** [12] as a representative decoder-modification baseline. The results are shown in the following table (The detailed experimental results can be found in **Supplementary Results: Comparison of SAC with Other Baseline Methods**.):
>
> ||ID|OOD|Avg|
> |-|-|-|-|
> |SAC|**54.95**|**39.26**|**47.11**|
> |Activation Beacon|40.31|36.91|38.61|
>
> SAC surpasses Activation Beacon by 8.5 points without decoder modifications, demonstrating the effectiveness of our approach.
>
> # Summary
>
> Thanks for your thoughtful and detailed review again. We hope our response addresses your comments. Please let us know if there are any additional questions, and we will be happy to discuss further!
>
> # Reference
>
> [1] Tao Ge, et al. In-context Autoencoder for Context Compression in a Large Language Model. ICLR, 2024.
>
> [2] Zongqian Li, et al. 500xCompressor: 500xCompressor: Generalized Prompt Compression for Large Language Models. ACL, 2025.
>
> [3] Runsong Zhao, et al. Position IDs Matter: An Enhanced Position Layout for Efficient Context Compression in Large Language Models. EMNLP Findings, 2025.
>
> [4] Ziyong Lin, et al. Look Both Ways and No Sink: Converting LLMs into Text Encoders without Training. ACL, 2025.
>
> [5] Chankyu Lee, et al. NV-Embed: Improved Techniques for Training LLMs as Generalist Embedding Models, ICLR, 2025.
>
> [6] Parishad BehnamGhader, et al. LLM2Vec: Large Language Models Are Secretly Powerful Text Encoders. COLM, 2024.
>
> [7] Zhenzhong Lan, et al. ALBERT: A Lite BERT for Self-supervised Learning of Language Representations. ICLR, 2020.
>
> [8] Yen, Howard. Long-Context Language Modeling with Parallel Context Encoding. MS thesis. Princeton University, 2024.
>
> [9] Li, Yuhong, et al. "SnapKV: LLM Knows What You are Looking for Before Generation." Advances in Neural Information Processing Systems 37 (2024): 22947-22970.
>
> [10] Zhang, Zhenyu, et al. "H2O: Heavy-Hitter Oracle for Efficient Generative Inference of Large Language Models." Advances in Neural Information Processing Systems 36 (2023): 34661-34710.
>
> [11] Han, Wei, et al. "Two are better than one: Context window extension with multi-grained self-injection." arXiv preprint arXiv:2410.19318 (2024).
>
> [12] Zhang, Peitian, et al. "Long Context Compression with Activation Beacon." arXiv preprint arXiv:2401.03462 (2024).

---

### Official Review · Reviewer_wACd · 2025-10-29

**Soundness:** 3
**Presentation:** 3
**Contribution:** 2
**Rating:** 6
**Confidence:** 4

**Summary:**

This paper introduces **Semantic-Anchor Compression (SAC)**, a new training-based method for LLM context compression. Its generation pipeline follows the prevailing setup:

- one language model acts as the compression model to produce compressed tokens from a given context, and
- a (frozen) generation model consumes those compressed tokens to generate outputs for downstream tasks.

Prior approaches----ICAE, 500xCompressor, and EPL ----first **pretrain with an autoencoding (AE) objective** to reconstruct the original context from compressed tokens, then fine-tune by passing the compressed tokens to a frozen generator, optimizing the LM objective and, finally, the downstream (e.g., QA) objective. The authors argue these methods require new, randomly initialized compression tokens*whose embeddings must be trained----on top of LoRA fine-tuning of the compression model----which "lack inherent semantic information" and thus demand extensive AE pretraining.

SAC addresses this by selecting anchor tokens directly from the original context, removing the need for AE pretraining and retaining only LM and task-level fine-tuning. Concretely, the context is partitioned into equal-length chunks, and the **center token** of each chunk is chosen as the **anchor** (following EPL). A fixed **anchor embedding** is then added to these tokens to highlight their special role. Finally, **bidirectional attention** is enabled **only in the compression model**, allowing each anchor to aggregate information from the **entire** context; the resulting **compressed KV states**----one per anchor----serve as the compact representation passed to the generation model. In experiments, the authors use Llama-3.2-1B for both compression and generation; during training, the compression model is fine-tuned with LoRA (r=128) while the generation model remains **frozen**.

To recap, SAC's contributions are twofold:
1. **Eliminates AE pretraining**, relying only on **LM** and **downstream (QA) fine-tuning**.
2. **Replaces learned compression tokens** with **semantically grounded anchor tokens** taken **from the original context**.

Two simple architectural tweaks make this effective: (i) add a dedicated **anchor embedding** to mark anchors, and (ii) apply **bidirectional attention** (in the compression model) so anchors can gather information from both preceding and succeeding tokens.

Empirically on MRQA, SAC consistently outperforms strong AE-based baselines (500xCompressor, EPL) on both in-domain and out-of-domain datasets, with the gap widening at higher compression ratios (e.g., 15× and 51×). The method also shows better training efficiency due to removing AE pretraining.

**Strengths:**

1. The critique of the AE's inefficiency is valid.

2. The core idea of using in-context "anchor tokens" is reasonable. It simplifies the compression training process and, as the t-SNE analysis (Fig. 5) suggests, produces compressed representations that are more aligned with the original token space.

3. SAC consistently outperforms baselines on MRQA, as well as during training (Appendix C trade-off curves). The fact that its performance advantage *increases* at higher compression ratios (15x, 51x) makes it more compelling. The experimental setups and compression ratios are consistent with prior works.

4. Table 5 shows that adding an AE task to the SAC architecture (SAC w/ AE+LM) actually *hurts* performance compared to the LM-only (SAC) model. This provides powerful evidence for the "AE mismatch" theory that the authors argue for, which is quite different from what the literature used to believe.

5. By removing the costly AE pretraining step, the method is inherently simpler and faster to train (as shown in Appendix B.1, Table 7).

**Weaknesses:**

1. The paper's default strategy (uniform chunking) is simple and effective. However, the ablation in Table 4 shows it performs similarly to a more complex method (Lingua-2). This suggests the selection strategy is important, but the paper stops short of exploring it deeply.

2. The baselines compared in the paper seem to be a bit outdated. The paper is missing an important baseline, DAST [1], which is also chunked compression but with a dynamic token budget, achieving decent performance at a high compression ratio.

3. The evaluation is performed exclusively on MRQA, a question-answering benchmark. This is a good choice, as QA is a key application. However, the paper's central critique is that the AE task is misaligned with *downstream tasks*. For a task like long-document summarization, an AE-style reconstruction objective might be *more* aligned than a QA objective. It is unclear if SAC's superiority would hold on tasks that require a more holistic understanding of the full context, rather than just extracting specific facts.

4. The paper states it "modifies the standard causal attention to a bidirectional attention mechanism." For clarity, it would be beneficial to explicitly state whether this bidirectional mask is applied *only* to the anchor tokens (i.e., their queries can see all keys/values) or if the entire compressor LLM (with LoRA) is run in a fully non-causal (BERT-style) mode during the compression pass.

5. All experiments are conducted on a single model at a single parameter size. It does not guarantee that the performance is generalizable to other models or models with much larger scales.

**Reference**:
Shaoshen Chen, Yangning Li, Zishan Xu, Yongqin Zeng, Shunlong Wu, Xinshuo Hu, Zifei Shan, Xin Su, Jiwei Tang, Yinghui Li, and Hai-Tao Zheng. 2025. DAST: Context-Aware Compression in LLMs via Dynamic Allocation of Soft Tokens. In Findings of the Association for Computational Linguistics: ACL 2025, pages 20544–20552, Vienna, Austria. Association for Computational Linguistics.

**Questions:**

1. (Regarding Weakness 1) Have you experimented with or considered a *learned* selection mechanism? It seems plausible that a small, trained module could identify more semantically salient tokens to serve as anchors, potentially improving performance further, especially at very high compression ratios. In addition, attention/entropy-based KV selection (i.e., from the KV eviction literature) could also be worth a try.

2. (Regarding Weakness 2) Can you add DAST[1] for comparison? Better if PCC [2] can also be included.

3. (Regarding Weakness 3) How do you anticipate SAC would perform on non-extractive, non-QA tasks like long-document summarization? The AE objective, while mismatched for QA, does force a holistic reconstruction. Does the LM-only pretraining capture enough global information for tasks that require a broader, generative understanding of the entire context?

3. (Regarding Weakness 4) Could you please clarify the precise implementation of the bidirectional attention? Is the attention mask modified only for the anchor token queries, or is the entire encoder pass non-causal for all tokens?

4. It would be better if more compression ratios could be experimented with, producing a trade-off curve between performance and compression ratio to further validate the robustness of SAC.

5. Can you clarify how you initialize the anchor embeddings?

**Minor Issues (does not affect evaluation)**:

1. At line 198, "While using original tokens … limiting its representation power." should be 1 sentence instead of 2.
2. At line 429, there should be a space within "Analysis.In".

**Reference**:
1. Shaoshen Chen, Yangning Li, Zishan Xu, Yongqin Zeng, Shunlong Wu, Xinshuo Hu, Zifei Shan, Xin Su, Jiwei Tang, Yinghui Li, and Hai-Tao Zheng. 2025. DAST: Context-Aware Compression in LLMs via Dynamic Allocation of Soft Tokens. In Findings of the Association for Computational Linguistics: ACL 2025, pages 20544–20552, Vienna, Austria. Association for Computational Linguistics.

2. Liao. 2025. Pretraining Context Compressor for Large Language Models with Embedding-Based Memory. In Proceedings of the 63rd Annual Meeting of the Association for Computational Linguistics (Volume 1: Long Papers), pages 28715–28732, Vienna, Austria. Association for Computational Linguistics.

---

> ### Author Response · Authors · 2025-11-21
> **Response by Authors to wACd (1/2)**
>
> Dear Reviewer wACd,
>
> We sincerely appreciate the time and effort you've devoted to reviewing our work and providing helpful feedback! In the following, we provide explanations to address your concerns.
>
> # Response to Weaknesses  & Questions
>
> ## For W1&Q1
>
> >  Have you experimented with or considered a *learned* selection mechanism? It seems plausible that a small, trained module could identify more semantically salient tokens to serve as anchors, potentially improving performance further, especially at very high compression ratios. In addition, attention/entropy-based KV selection (i.e., from the KV eviction literature) could also be worth a try.
>
> Thank you for the insightful suggestion regarding the learnable selection mechanism. We agree that this represents a valuable direction for exploration.
>
> Following your advice, we constructed a simple learnable module to perform a preliminary investigation. The specific experimental setup is detailed in **Supplementary Results: Anchor Selection Strategy Comparison for SAC**.
>
> ||ID|OOD|Avg|
> |-|-|-|-|
> |SAC|**54.95**|**39.26**|**47.11**|
> |SAC(learned selection)|52.10|38.06|45.08|
>
> The results show that incorporating the learnable module significantly degrades performance. We hypothesize two main reasons for this performance drop:
>
> - The model may struggle to spontaneously learn the distribution of important tokens solely from the implicit knowledge acquired during pre-training. A selection mechanism might require specific task-based training to be effective.
>
> - Compared to SAC (e.g., using a fixed/deterministic anchor) the uncertainty introduced by the selection mechanism may have made the optimization and learning process more challenging.
>
> We acknowledge that adopting selection mechanisms derived from the KV eviction literature to choose anchor tokens may indeed be effective. However, given the time constraints of the rebuttal period, we are unable to provide these experiments as scheduled. We commit to exploring this promising direction further in our future work.
>
> ## For W2&Q2
>
> > Can you add DAST[1] for comparison? Better if PCC [2] can also be included.
>
> We appreciate the reviewer’s suggestion. As the original DAST [1] codebase was unavailable, we re-implemented the method following the description in the paper and detailed the experiment setups in the **Supplementary Results: Comparison of SAC with Other Baseline Methods**
>
> ||ID|OOD|Avg|
> |-|-|-|-|
> |SAC|**54.95**|**39.26**|**47.11**|
> |DAST|49.83|31.83|40.83|
>
> The results show that SAC consistently outperforms DAST in both IID and OOD scenarios, achieving improvements of 10.27% and 23.34%, respectively. These findings further validate the effectiveness of the SAC architecture.
>
> ## For W3&Q3
>
> > How do you anticipate SAC would perform on non-extractive, non-QA tasks like long-document summarization? The AE objective, while mismatched for QA, does force a holistic reconstruction. Does the LM-only pretraining capture enough global information for tasks that require a broader, generative understanding of the entire context?
>
> We thank the reviewer for the valuable suggestion. We agree that the AE objective may be more beneficial for tasks requiring global information. To investigate this, we conducted experiments on two long-document summarization datasets, QMSum [2] and GovReport [3], using a maximum sequence length of 32K and a compression rate of 15×.
>
> ||qmsum|govreport|Summ avg|
> |-|-|-|-|
> |SAC|14.95|**22.03**|**18.49**|
> |SAC(ae+lm)|**15.32**|20.15|17.74|
> |EPL|14.82|20.4|17.61|
>
> The results show that SAC achieves the highest average performance on long-text summarization tasks. It is worth noting that QMSum has significantly less training data (1.26k) compared to GovReport (17.5k), which may explain why SAC(ae+lm) performs slightly better on QMSum while SAC excels on the larger GovReport dataset. We conjecture that as the amount of redundant information increases in long sequences and the information capacity of compression tokens has an upper bound, the AE objective's requirement to reconstruct all information may impose an additional burden on the model, thereby negatively impacting performance.

---

> ### Author Response · Authors · 2025-11-21
> **Response by Authors to wACd (2/2)**
>
> ## For W4&Q4
>
> > Could you please clarify the precise implementation of the bidirectional attention? Is the attention mask modified only for the anchor token queries, or is the entire encoder pass non-causal for all tokens?
>
> we appreciate the reviewer’s suggestion. We will further elaborate on the implementation details and incorporate them into our revised version.
>
>
> **The entire encoder pass is non-causal for all tokens.** This design choice was primarily inspired by recent successful practices in the fields of retrieval and representation learning. For example, works such as NV-Embed [4] and LLM2Vec [5] have shown that converting an autoregressive LLM into a bidirectional attention structure can effectively enhance the model’s representation capability. Our ablation study in Section 4.3 further validates the effectiveness of bidirectional attention.
>
> ## For W5&Q5
>
> > It would be better if more compression ratios could be experimented with, producing a trade-off curve between performance and compression ratio to further validate the robustness of SAC.
>
> Thank you for your suggestion. We are currently training experiments at additional compression ratios to obtain a trade-off curve between generation performance and compression rate, in order to further verify the robustness of SAC. Due to time constraints, we have only plotted the performance curves for the existing compression ratios (**This is shown in Figure 11 of our paper.**). We will update the results for more compression ratios as soon as they become available.
>
> ## For W6&Q6
>
> > Can you clarify how you initialize the anchor embeddings?
>
> We appreciate the reviewer’s suggestion and acknowledge that the description in the Methods section is not sufficiently clear. In our revised version, we will provide a more detailed explanation of the implementation of the anchor embedding.
>
> In our implementation, the anchor embedding initialization follows a normal distribution, where each element is randomly sampled from a distribution with a mean of 0 and a standard deviation of 0.02. The anchor embedding $e_A$ is a learnable vector with the same dimensionality as the original token embedding $E(c_i)$.
>
> # Summary
>
> We sincerely appreciate your thoughtful and detailed review. We hope our response addresses your comments. Please let us know if there are any additional questions, and we will be happy to discuss further!
>
> # Reference
> [1] Shaoshen Chen, et al. DAST: Context-Aware Compression in LLMs via Dynamic Allocation of Soft Tokens. ACL Findings, 2025.
>
> [2] Ming Zhong, et al. QMSum: A New Benchmark for Query-based Multi-domain Meeting Summarization. NAACL, 2021.
>
> [3] Luyang Huang, et al. Efficient Attentions for Long Document Summarization. NAACL, 2021.
>
> [4] Chankyu Lee, et al. NV-Embed: Improved Techniques for Training LLMs as Generalist Embedding Models, ICLR, 2025.
>
> [5] Parishad BehnamGhader, et al. LLM2Vec: Large Language Models Are Secretly Powerful Text Encoders. COLM, 2024.

---

> > ### Comment · Reviewer_wACd · 2025-11-25
> >
> > Thank you for responses. Raising score.

---

> > > ### Author Response · Authors · 2025-11-25
> > >
> > > We truly appreciate your positive comments. These valuable feedbacks will be fully incorporated into the next version.
> > >
> > > Thanks for your efforts again！

---

### Official Review · Reviewer_NhSy · 2025-10-31

**Soundness:** 3
**Presentation:** 3
**Contribution:** 3
**Rating:** 6
**Confidence:** 4

**Summary:**

This paper proposes Semantic-Anchor Compression, a novel framework designed to improve the efficiency and performance of Large Language Models on long-context tasks. The method introduces an autoencoding-free approach to context compression. The core novelty lies in three architectural designs: 1) It selects 'anchor tokens' directly from the original context to act as information carriers. 2) It introduces a trainable 'anchor embedding' to semantically distinguish these selected tokens. 3) It enables 'bidirectional attention' specifically for these anchor tokens, allowing them to effectively aggregate global context. This design allows the compression mechanism to be optimized directly for downstream tasks. Experimental results on a benchmark dataset demonstrate that the method achieves strong performance across various compression ratios and shows favorable generalization capabilities.

**Strengths:**

1. The paper introduces a novel and effective information compression method based on anchor tokens. This "autoencoding-free" paradigm successfully moves beyond traditional context-agnostic compression tokens.

2. The proposed SAC method is simple, architecturally concise, and elegant in its design.

3. The paper is exceptionally well-written and well-structured, clearly articulating the core problem, the proposed solution, and its advantages.

**Weaknesses:**

1. The paper's default strategy is to select the "middle token" of a chunk. However, the ablations (Table 4) are missing a critical and intuitive baseline: selecting the "last token" of each chunk as the anchor. In a causal model, the last token has naturally seen the entire chunk. This is a significant omission that prevents a full assessment of the "middle token" strategy and makes it difficult to judge if the performance gain comes from a truly better mechanism or just a better-positioned token.
2. All experiments are conducted only on QA tasks. As the authors state, the AE task deviates from "real-world usage," but using only QA to represent "real-world usage" is far from sufficient. SAC is a lossy compression (as it discards the AE objective), making it highly likely to have overfitted to the specific patterns of QA tasks. Its performance on other long-context tasks (e.g., summarization, story generation) is unknown.

3. The paper's core argument is that AE-free is better. However, the authors never test the SAC + AE combination. That is, using SAC's anchor architecture (with bidirectional attention) to simultaneously perform the downstream task and the AE reconstruction task. It is plausible that SAC + AE would have slightly lower QA performance but be more robust and general-purpose (by retaining reconstruction). The authors claim AE is harmful but do not prove this experimentally (i.e., by comparing SAC vs. SAC + AE).

**Questions:**

1. Can you please provide a baseline experiment using the "last token" of each chunk as the anchor, while still applying your anchor embedding and bidirectional attention? I am very curious how this intuitive baseline compares to your "middle token" strategy.

2. Do you have plans to validate SAC on other long-context tasks besides QA (e.g., summarization, multi-turn dialogue)? The current setup makes me highly concerned that the model is overfitting to QA while sacrificing general compression capabilities.

3. Did you consider testing a SAC + AE setup (i.e., using your anchor architecture but retaining the AE loss)? What are your thoughts on the trade-off between QA performance and general-purpose robustness that this combination might offer?

4. Could you provide a detailed analysis of the inference overhead? While SAC saves on AE training, what is the specific extra compute (FLOPs) or latency cost of enabling bidirectional attention for anchor tokens (either over the full sequence or within chunks) during inference, compared to standard KV caching or older compression methods?

---

> ### Author Response · Authors · 2025-11-21
> **Response by Authors to NhSy (1/2)**
>
> Dear Reviewer NhSy,
>
> We sincerely thank you for the valuable feedback. To further improve the quality of our paper, we have included additional experiments as suggested.
>
> # Response to Weaknesses  & Questions
>
> ## For W1&Q1
>
> > Can you please provide a baseline experiment using the "last token" of each chunk as the anchor, while still applying your anchor embedding and bidirectional attention? I am very curious how this intuitive baseline compares to your "middle token" strategy.
>
> We thank you for raising this insightful question. In our main submission, we adopt the middle-token anchor as the default setting. This choice is motivated by prior work [1], which shows that for context-compression settings, placing the compress token near the center of a chunk yields more balanced positional encoding and consistently better empirical performance than alternative positional choices.
>
> That said, we fully agree with you that evaluating a last-token anchor is an important and intuitive baseline—particularly because, in causal models, the last token naturally attends to the entire chunk. To address this concern, we conducted **additional experiments** using the last token of each chunk as the anchor while keeping the anchor embedding and bidirectional attention unchanged. The evaluation results are shown in the table below  (The detailed experimental results can be found in  **Supplementary Results: Anchor Selection Strategy Comparison for SAC**.):
>
> ||ID|OOD|Avg|
> |-|-|-|-|
> |SAC|**54.95**|**39.26**|**47.11**|
> |SAC(chunk last token)|54.09|38.71|46.40|
> |EPL|51.52|36.74|44.13|
>
> The results show that using the middle token of each chunk consistently outperforms using the last token, which aligns with the insight reported in [1]. At the same time, the last-token anchor still performs better than the other baselines, indicating that the improvements do not merely come from positional advantages but from the effectiveness of our proposed anchoring mechanism.
>
> ## For W2&Q2
>
> > Do you have plans to validate SAC on other long-context tasks besides QA (e.g., summarization, multi-turn dialogue)? The current setup makes me highly concerned that the model is overfitting to QA while sacrificing general compression capabilities.
>
> Thank you for raising this important concern regarding SAC potentially overfitting to QA tasks due to the removal of the AE objective. We fully agree that evaluating SAC on broader long-context tasks beyond QA is valuable. In the initial version of the paper, we focused on QA tasks partly because this follows prior work and partly because QA provides a controlled and standardized evaluation setting.
>
> However, we also recognize that QA alone does not capture the full spectrum of real-world long-context applications. To address this limitation, we have conducted additional experiments on long-text summarization tasks, specifically QMSum [2] and GovReport [3]. We trained SAC and EPL on the corresponding training sets using a maximum input length of 32K tokens and a compression ratio of 15×. The results are presented in the table below (The detailed experimental results can be found in **Supplementary Results: Long-Context Task Evaluation Results**.):
>
> ||Sigle-doc QA|Multi-doc QA|Summ|Avg|
> |-|-|-|-|-|
> |SAC|**13.58**|**21.06**|**18.49**|**17.71**|
> |EPL|10.81|18.05|17.61|15.49|
>
> The results show that SAC maintains strong performance on these non-QA tasks, indicating that removing the AE objective does not harm the model’s general long-context modeling capability. We appreciate your suggestion, which has helped strengthen the paper, and we hope these results alleviate the concern about QA-specific overfitting. We will include these additional experimental results in the revised version.

---

> ### Author Response · Authors · 2025-11-21
> **Response by Authors to NhSy (2/2)**
>
> ## For W3&Q3
>
>
> > Did you consider testing a SAC + AE setup (i.e., using your anchor architecture but retaining the AE loss)? What are your thoughts on the trade-off between QA performance and general-purpose robustness that this combination might offer?
>
> Thank you for pointing out the importance of evaluating SAC with the AE loss retained. We apologize for the misunderstanding — in fact, our paper already includes this baseline in Table 5 (SAC w/ AE + LM). The results show that incorporating the AE objective indeed leads to a clear drop in QA performance.
>
> To further investigate whether this degradation is QA-specific or reflects a broader conflict between AE and downstream tasks, we additionally evaluated SAC vs. SAC w/ AE on other long-context tasks. The results are summarized in the table below (The detailed experimental results can be found in **Supplementary Results: Long-Context Task Evaluation Results**.):
>
> ||Sigle-doc QA|Multi-doc QA|Summ|Avg|
> |-|-|-|-|-|
> |SAC|**13.58**|**21.06**|**18.49**|**17.71**|
> |SAC(ae+lm)|12.31|18.04|17.74|16.03|
>
> These experiments demonstrate that removing the AE objective not only benefits QA performance but also overall generalizes well across non-QA tasks (with minor degradation on qmsum). This suggests that the conflict between AE and downstream objectives is not limited to QA, and further supports the rationale behind our SAC design.
>
> ## For W4&Q4
>
> > Could you provide a detailed analysis of the inference overhead? While SAC saves on AE training, what is the specific extra compute (FLOPs) or latency cost of enabling bidirectional attention for anchor tokens (either over the full sequence or within chunks) during inference, compared to standard KV caching or older compression methods?
>
> Thank you for this important question. We want to clarify a critical advantage of SAC: while SAC does enable bidirectional attention for anchor tokens, it does **not** require appending additional $k$ special tokens to the sequence like 500xCompressor. This means **SAC operates on shorter sequences** during inference, directly reducing computational overhead.
>
> Let's quantify this advantage. Given a context with shape $[b, s, h]$ where $b$ is batch size, $s$ is sequence length, $h$ is hidden size, and $I$ is the FFN intermediate size, we compare the theoretical FLOPs:
>
> | **Modules**                 | **SAC-FLOPs**    | **500x-FLOPs**          |
> | --------------------------- | ---------------- | ----------------------- |
> | $\mathbf{x(W_Q/W_K/W_V)}$   | $3 \cdot 2bsh^2$ | $3 \cdot 2b(s+k)h^2$    |
> | $\mathbf{QK^T}$             | $2bs^2h$         | $b(s+k)^2h$             |
> | $\mathbf{AV}$               | $2bs^2h$         | $b(s+k)^2h$             |
> | $\mathbf{xW_O}$             | $2bsh^2$         | $2b(s+k)h^2$            |
> | $\mathbf{X_{out} W_{up}}$   | $2bshI$          | $2b(s+k)hI$             |
> | $\mathbf{X_{out} W_{gate}}$ | $2bshI$          | $2b(s+k)hI$             |
> | $\mathbf{X_{out} W_{down}}$ | $2bshI$          | $2b(s+k)hI$             |
> | $\mathbf{sum}$              | $bhs(8h+4s+6I)$  | $bh(s+k)[8h+2(s+k)+6I]$ |
>
> **Concrete comparison**: With typical settings ($b=1, s=510, h=2048, I=8192$):
> - At 5× compression ($k=102$): 500xCompressor requires **1.19× more FLOPs** than SAC
> - At 10× compression ($k=51$): 500xCompressor requires **1.08× more FLOPs** than SAC
>
> Despite using bidirectional attention, SAC is actually more efficient than 500xCompressor because it doesn't inflate the sequence length. The bidirectional attention cost is more than offset by operating on the original, shorter sequence.
>
> # Summary
>
> We sincerely appreciate your thoughtful and detailed review. We hope our response addresses your comments. Please let us know if there are any additional questions, and we will be happy to discuss further!
>
> # Reference
>
> [1] Runsong Zhao, et al. Position IDs Matter: An Enhanced Position Layout for Efficient Context Compression in Large Language Models. EMNLP Findings, 2025.
>
> [2] Ming Zhong, et al. QMSum: A New Benchmark for Query-based Multi-domain Meeting Summarization. NAACL, 2021.
>
> [3] Luyang Huang, et al. Efficient Attentions for Long Document Summarization. NAACL, 2021.

---

### Official Review · Reviewer_rbBK · 2025-11-01

**Soundness:** 2
**Presentation:** 2
**Contribution:** 2
**Rating:** 4
**Confidence:** 3

**Summary:**

This paper proposes a context compression method that does not relying on autoencoding training. The proposed method, SAC, selects anchor tokens directly from the original tokens and aggregates contextual information into their key-value tokens using bidrectional attention. Experiments show that SAC outperforms existing context compression methods.

**Strengths:**

1. Context compression is an important problem worthy of investigation.
2. The proposed method is simple. It combines token selection, common in KV cache compression but probably not in context compression, and bidirectional attention.
3. The proposed method outperfoms existing context compression methods.

**Weaknesses:**

1. The method is not clearly described and sometimes very confusing.
2. The contexts in the experiments are in general short. It is of more interest to compress long contexts.
3. The experiments use a single model, Llama-3.2-1B. It is unclear whether the proposed method generalizes well to larger models and other model families.
4. The paper has an observation that challenges previous work, but does not provide any explanation, so it is hard to see if is a bug or something new.

**Questions:**

1. Can you give a clear description of the method? For example, what are the anchor embeddings? How does anchor embeddings enable the compressor to identify critical tokens? The anchor tokens are just sampled evenly, so why are they more important? Why do you choose the output of anchor tokens from the LLM's final layers or the KV pairs from each layer? Any justifications?
2. Can you demonstrate the effectiveness and performance improvement of the proposed method for long contexts?
3. Can you provide evidence for the generalizability of the proposed model to larger models and other model families?
4. What do you think might have caused the discrepancy in the observed AE effect?

---

> ### Author Response · Authors · 2025-11-21
> **Response by Authors to rbBK (1/2)**
>
> Dear Reviewer rbBK,
>
> Thank you for your constructive review. We appreciate your recognition of the importance of context compression and your acknowledgment of SAC's simplicity and strong experimental performance. Below, we address your concerns and provide additional experimental results.
>
> # Response to Weaknesses and Questions
>
> ## For W1&Q1
> > Can you give a clear description of the method?
>
> Yes. The SAC method can be summarized into the following four steps:
>
> - **Anchor Selection**: A subset of tokens $S$ are selected from the original context $C$ to serve as the semantic anchors. Uniform sampling is adopted by default.
>
> - **Anchor Enhancement**: We add a learnable anchor embedding vector $e_A$ to the embeddings of $S$ to explicitly mark these tokens as semantic anchors.
>
>
> - **Compression**: A modified compressor whose architecture is the same as the target LLM with LoRA adapters processes the full context $C$ with bidirectional attention, where anchor tokens $S$ are marked by $e_A$. The compressor then aggregates information and outputs the compressed KV pairs of anchors.
>
> - **LLM Inference**: The compressed KV pairs serve as the KV cache for the target LLM, which conditions on them together with the query to generate responses.
>
> > What are the anchor embeddings?
>
> The anchor embedding $e_A \in \mathbb{R}^d$ is a learnable parameter vector that shares the same dimension $d$ as the original token embeddings $E(c_i)$. This embedding is added to the embeddings of anchors to provide an explicit signal indicating which tokens serve as semantic anchors during context compression.
>
> > How does anchor embeddings enable the compressor to identify critical tokens?
>
> We clarify that the function of the anchor embedding $e_A$ is not to identify critical tokens. Rather, the anchors are pre-selected (e.g., via uniform sampling). The anchor embedding serves as marker that signals to the compressor which tokens should aggregate contextual information during the compression process.
>
> > The anchor tokens are just sampled evenly, so why are they more important?
>
> Based on the empirical results in our submitted Table 4 and the supplementary results “Anchor Selection Strategy Comparison for SAC”, the semantic content of the anchor tokens does not appear to be important. Rather, they serve as compression carriers. Due to the properties of Rotary Position Embeddings (RoPE[1]), anchors tend to aggregate information from their surrounding tokens, as evidenced by the attention patterns in Figure 7 and [2]. Uniform sampling ensures that these compression carriers are evenly distributed across the full context, maximizing coverage of the entire sequence.
>
> > Why do you choose the output of anchor tokens from the LLM's final layers or the KV pairs from each layer?
>
> Following [2,3], we extract KV pairs from each layer for direct plug-and-play insertion into the LLM's KV cache. According to [3], this design significantly improves performance compared to using final-layer outputs as in ICAE [4].
>
> ## For W2&Q2
> > Can you demonstrate the effectiveness and performance improvement of the proposed method for long contexts?
>
> To directly address this question, we conducted an experiment evaluating SAC's effectiveness on contexts significantly longer than those seen during training. We test the resulting SAC and EPL model (trained with a 15× compression ratio on 2K-token contexts) on a 24K-token long-context QA benchmark (The detailed experimental results can be found in  **Supplementary Results: Long-Context Task Evaluation Results**.):
>
> ||Sigle-doc QA|Multi-doc QA|Avg|
> |-|-|-|-|
> |SAC|**13.58**|**21.06**|**17.32**|
> |EPL|10.81|18.05|14.43|
>
> SAC consistently outperforms the best baseline compression methods at a 15x compression ratio in both single-document and multi-document QA tasks. This demonstrates that our method maintains its performance advantage even when applied to much longer sequences.

---

> ### Author Response · Authors · 2025-11-21
> **Response by Authors to rbBK (2/2)**
>
> ## For W3&Q3
>
> > Can you provide evidence for the generalizability of the proposed model to larger models and other model families?
>
> To evaluate the generalizability of SAC across model scales, we have conducted additional experiments on the larger LLaMA 3B and 8B models. The results, detailed in the provided table, strongly support its scalability  (The detailed experimental results can be found in  **Supplementary Results: Scale-up Evaluation Results**.):
>
> ||ID|OOD|Avg|
> |-|-|-|-|
> |SAC(3B)|**65.12**|**50.48**|**57.80**|
> |EPL(3B)|63.95|47.46|55.71|
> |SAC(8B)|**67.09**|**52.31**|**59.70**|
> |EPL(8B)|67.08|50.82|58.95|
>
>
> * On the 3B model, SAC achieves an average ROUGE-1 score of 50.48 and an Exact Match (EM) score of 34.73, outperforming the EPL baseline (47.46 / 31.82).
>
> * On the 8B model, SAC scores 52.31 (ROUGE-1) and 35.93 (EM), again surpassing the EPL baseline (50.82 / 34.42).
>
> These consistent gains demonstrate that SAC effectively generalizes to larger model sizes and maintains its performance advantage.
>
> ## For W4&Q4
> > What do you think might have caused the discrepancy in the observed AE effect?
>
> We believe the reasons that may lead to the difference in the AE effect mainly include the following two points:
> * Intuitive Perspective: Low-information-density tokens inevitably exist in the sequence, yet AE loss requires the compressed representations to reconstruct the entire context. This places a burden on the model's compression representation capability, potentially causing the model to use its limited capacity to encode redundant details.
> * AE-LM Gradient Conflict: As shown in the table below, joint training with the LM task degrades AE reconstruction performance (99.94 to 98.19), particularly at high compression ratios.  We attribute this to gradient misalignment between the two objectives.  Our visualization analysis in Figure 10 confirms that the gradient cosine similarity between AE and LM losses approaches zero during training.  This indicates that the two tasks are largely orthogonal in parameter space, causing the optimization of AE to hinder LM updates and downstream performance.
>
> ||AE(ppl)|LM(ppl)|ae_bleu4|
> |-|-|-|-|
> |ICAE|4.08|12.35|12.02|
> |500x|1.09|11.83|87.21|
> |EPL|1.03|10.88|97.50|
> |SAC(ae+lm)|1.03|11.01|98.19|
> |SAC(only ae)|**1.00**|-|**99.94**|
> |SAC|-|**10.79**|-|
>
> # Summary
>
> We hope the above clarifications and additional results have fully addressed your concerns. We would be very grateful if you would consider these points in your final assessment of our contribution.
>
> # Reference
> [1] Jianlin Su, et al. “RoFormer: Enhanced Transformer with Rotary Position Embedding.” arXiv preprint arXiv:2104.09864, 2023.
>
> [2] Runsong Zhao, et al. Position IDs Matter: An Enhanced Position Layout for Efficient Context Compression in Large Language Models. EMNLP Findings, 2025.
>
> [3] Zongqian Li, et al. 500xCompressor: 500xCompressor: Generalized Prompt Compression for Large Language Models. ACL, 2025.
>
> [4] Tao Ge, et al. In-context Autoencoder for Context Compression in a Large Language Model. ICLR, 2024.

---

### Author Response · Authors · 2025-11-21
**Supplementary Results - part 1/2**

We sincerely thank all the reviewers for their constructive suggestions. In response, we have carefully analyzed their concerns and added the following experimental results. Except for the larger-scale experiments, all our experiments are conducted using Llama-3.2-1B with a fixed 15 times compression rate, and the evaluation metric is ROUGE-1 F1.To ensure the fairness of the comparison, all our comparative experiments maintained the same training data and hyperparameter settings.

# Long-Context Task Evaluation Results (Summarization & QA)

We are grateful to the reviewers for their concern about the performance of SAC in long text tasks.
For the summarization tasks, we trained SAC and EPL [1] on QMSum [2] and GovReport [3] with a maximum input length of 32K tokens, and their test results are shown in Table 1. Meanwhile, We also perfrom the test on a 24k token-long context QA benchmark, using the model checkpoint following the settings in the paper trained with a 15x compression ratio in a 2k token context, and the results are shown in Table 2.

Table 1: Model Performance on 32K Long-Text Summarization Tasks.
||QMSum|GovReport|Summ Avg|
|:-:|:-:|:-:|:-:|
|SAC|14.95|**22.03**|**18.49**|
|SAC(ae+lm)|**15.32**|20.15|17.74|
|EPL|14.82|20.40|17.61|

Table 2: Model Performance on 24K Long-Text QA Tasks.
||MultifieldQA|NarrativeQA|Qasper|Sigle-doc QA Avg|2Wikimqa|Musique|HotpotQA|Multi-doc QA Avg|
|:-:|:-:|:-:|:-:|:-:|:-:|:-:|:-:|:-:|
|SAC|**17.75**|**11.33**|**11.67**|**13.58**|**24.78**|**12.36**|**26.03**|**21.06**|
|SAC(ae+lm)|17.38|10.49|9.05|12.31|22.47|7.76|23.88|18.04|
|EPL|14.34|10.50|7.58|10.81|21.99|8.43|23.72|18.05|

# Scale-up Evaluation Results

We thank the reviewers for their concern regarding larger-scale experiments. We trained Llama3.2-3B and Llama-3.1-8B on 8 A100 GPUs using the same data and training settings as described in the paper.

Table 3: Experimental Results of the 1B/3B/8B Models on the In-Domain QA Tasks.

||SQuAD|NewsQA|TriviaQA|SearchQA|HotpotQA|NQ|Avg|
|:-:|:-:|:-:|:-:|:-:|:-:|:-:|:-:|
|SAC(1B)|**47.43**|**36.55**|**61.13**|**68.97**|**58.83**|**56.79**|**54.95**|
|EPL(1B)|44.58|33.34|56.16|66.36|54.88|53.80|51.52|
|SAC(3B)|**63.11**|**49.23**|**69.65**|**73.88**|**68.87**|**65.98**|**65.12**|
|EPL(3B)|62.43|46.72|68.83|73.61|68.50|63.59|63.95|
|SAC(8B)|64.92|49.77|72.37|**76.92**|70.80|**67.77**|**67.09**|
|EPL(8B)|**65.79**|**51.18**|**72.4**|74.86|**70.98**|67.24|67.08|

Table 4: Experimental Results of the 1B/3B/8B Models on the Out-of-Domain QA Tasks.

||BioASQ|DROP|DuoRC|RACE|RE|TextbookQA|Avg|
|:-:|:-:|:-:|:-:|:-:|:-:|:-:|:-:|
|SAC(1B)|**41.31**|**36.72**|**28.94**|**23.35**|**61.04**|**44.21**|**39.26**|
|EPL(1B)|40.52|32.16|25.70|20.97|59.75|41.31|36.74|
|SAC(3B)|**47.96**|**48.46**|**42.21**|33.61|**75.11**|**55.53**|**50.48**|
|EPL(3B)|46.76|46.20|36.81|**33.80**|65.73|55.46|47.46|
|SAC(8B)|48.54|**51.55**|**41.44**|**35.52**|**78.91**|**57.87**|**52.31**|
|EPL(8B)|**52.52**|49.98|40.09|35.22|69.77|57.36|50.82|

# Anchor Selection Strategy Comparison for SAC

We thank the reviewer for this insightful suggestion. We have experimented with a learned selection mechanism as follows. To enable end-to-end training with discrete Top-K selection, we employ a differentiable approximation: the input is first mapped to logits via a linear layer, then relaxed into a continuous probability distribution using Gumbel-Softmax. During the forward pass, we perform hard Top-K selection to identify anchor tokens, while during backpropagation, we apply the straight-through (ST) estimator—gradients bypass the non-differentiable Top-K operation and flow through the soft distribution to update the linear layer. This allows the module to automatically learn which tokens best preserve sequence semantics and minimize the final LM loss.

Table 5: Performance of Anchor Selection Strategies on In-Domain QA Tasks.

||SQuAD|NewsQA|TriviaQA|SearchQA|HotpotQA|NQ|Avg|
|:-:|:-:|:-:|:-:|:-:|:-:|:-:|:-:|
|SAC|**47.43**|**36.55**|**61.13**|**68.97**|**58.83**|**56.79**|**54.95**|
|SAC(chunk last token)|46.59|36.03|60.80|67.20|58.51|55.41|54.09|
|SAC(learned selection)|42.32|33.69|59.04|68.27|55.52|53.76|52.10|

Table 6: Performance of Anchor Selection Strategies on Out-of-Domain QA Tasks

||BioASQ|DROP|DuoRC|RACE|RE|TextbookQA|Avg|
|:-:|:-:|:-:|:-:|:-:|:-:|:-:|:-:|
|SAC|41.31|**36.72**|28.94|23.35|**61.04**|44.21|**39.26**|
|SAC(chunk last token)|**42.87**|35.61|**29.11**|**23.39**|56.11| **45.17**|38.71|
|SAC(learned selection)|41.93|34.12|28.64|21.07|58.24|44.38|38.06|

---

### Author Response · Authors · 2025-11-21
**Supplementary Results - part 2/2**

# Comparison of SAC with Other Baseline Methods

We thank the reviewers for suggesting the comparative baseline, which has enriched our paper. Since the DAST [4] codebase is empty, we reproduced it according to the descriptions in the paper. For a fair comparison, all hyperparameter settings and training data are identical to those used for SAC.

For Activation Beacon [6], we employed the same training data and only modified some hyperparameter settings to ensure alignment with SAC, specifically setting beacon_window=510 and beacon_ratio=[2,4,8,15], while keeping the remaining settings consistent with the examples in the codebase.

Table 7: Baseline Performance Comparison on In-Domain QA Tasks.

||SQuAD|NewsQA|TriviaQA|SearchQA|HotpotQA|NQ|Avg|
|:-:|:-:|:-:|:-:|:-:|:-:|:-:|:-:|
|SAC|**47.43**|**36.55**|**61.13**|**68.97**|**58.83**|**56.79**|**54.95**|
|DAST|36.33|31.55|56.92|68.07|54.02|52.10|49.83|
|Activation Beacon|37.53|31.10|48.85|39.06|45.01|40.29|40.31|

Table 8: Baseline Performance Comparison on Out-of-Domain QA Tasks.

||BioASQ|DROP|DuoRC|RACE|RE|TextbookQA|Avg|
|:-:|:-:|:-:|:-:|:-:|:-:|:-:|:-:|
|SAC|**41.31**|**36.72**|28.94|23.35|**61.04**|**44.21**|**39.26**|
|DAST|36.57|31.90|21.56|16.31|48.54|36.69|31.83|
|Activation Beacon|36.07|34.39|**33.78**|**26.68**|53.28|37.27|36.91|

# Effect of Different Compressors

We thank the reviewer for the insightful suggestion.  Since ALBERT's [5] KV layers cannot align with the decoder, we chose to use the hidden state as the compressed representation and compare with ICAE which also uses hidden state as the compressed representation.  For fair comparison, we maintained identical training data and hyperparameter settings, with the results shown in the table below.

Table 9: Performance of Different Compressors on In-Domain Tasks.

||SQuAD|NewsQA|TriviaQA|SearchQA|HotpotQA|NQ|Avg|
|:-:|:-:|:-:|:-:|:-:|:-:|:-:|:-:|
|SAC|**47.43**|**36.55**|**61.13**|**68.97**|**58.83**|**56.79**|**54.95**|
|ICAE-Llama|31.90|25.25|51.78|64.81|45.22|48.01|44.50|
|ICAE-ALBert|13.16|6.47|19.13|30.93|12.09|13.06|15.81|

Table 10: Performance of Different Compressors on Out-of-Domain Tasks.

||BioASQ|DROP|DuoRC|RACE|RE|TextbookQA|Avg|
|:-:|:-:|:-:|:-:|:-:|:-:|:-:|:-:|
|SAC|**41.31**|**36.72**|**28.94**|**23.35**|**61.04**|**44.21**|**39.26**|
|ICAE-Llama|35.51|30.39|13.78|15.21|55.24|34.75|30.81|
|ICAE-ALBert|23.73|21.62|3.37|5.56|15.62|25.15|15.84|

# Reference

[1] Runsong Zhao, et al. Position IDs Matter: An Enhanced Position Layout for Efficient Context Compression in Large Language Models. EMNLP Findings, 2025.

[2] Ming Zhong, et al. QMSum: A New Benchmark for Query-based Multi-domain Meeting Summarization. NAACL, 2021.

[3] Luyang Huang, et al. Efficient Attentions for Long Document Summarization. NAACL, 2021.

[4] Shaoshen Chen, et al. DAST: Context-Aware Compression in LLMs via Dynamic Allocation of Soft Tokens. ACL Findings, 2025.

[5] Zhenzhong Lan, et al. ALBERT: A Lite BERT for Self-supervised Learning of Language Representations. ICLR, 2020.

[6] Zhang, Peitian, et al. "Long Context Compression with Activation Beacon." arXiv preprint arXiv:2401.03462 (2024).

---

### Comment · Area_Chair_tMGU · 2025-11-24
**Rebuttal Received - Next Steps**

Dear Reviewers,

The authors have submitted their rebuttal. Please review their responses and provide any follow-up, such as additional questions or revisions to your review.

As the scores split equally into positive and negative views (two positives (score 6) and two negatives (score 4 & 2)), I kindly encourage you to carefully consider the other reviewers' perspectives and have more discussions to help us achieve a consensus.

Thank you for your contributions to this process.

Sincerely, Your AC

---

### Author Response · Authors · 2025-11-27

Dear Reviewers,

We sincerely appreciate the time and effort you have dedicated to reviewing our work. We understand that your schedule is busy, and we are truly grateful for your valuable feedback. As the author–reviewer discussion period is now halfway through,  we would be very grateful for the opportunity to have a further dialogue. Our aim is to gain insights into whether our responses effectively address your concerns and to ascertain if there are any additional questions or points you would like to discuss.

We look forward to the opportunity for further discussion with you. Thank you for your thoughtful consideration.

Best regards, Authors

---

### Author Response · Authors · 2025-12-01
**Consolidated Summary of the Paper and Rebuttal Progress**

Dear Area Chair,

Thank you for handling this submission under special circumstances, and we also sincerely appreciate the reviewers for their time and constructive feedback. Below, we would like to summarize our paper as well as the discussions with reviewers so far, hopefully can help to facilitate the whole reviewing process.

##  I. Core Contributions Generally Acknowledged by Reviewers

This paper aims to address the limitations of current context compression methods and proposes improvements to enhance their performance. Existing compression methods ([1][2][3][4]) rely on autoencoding (AE) to endow compression tokens with compression capability, but their reconstruction objectives do not align with real downstream tasks, thereby weakening features that would be more useful in practical applications. To address this issue, we propose Semantic Anchor Compression (SAC), a novel compression architecture that derives the compression representation based on context tokens directly; SAC shows to achieve superior performance without AE. Among all reviewer comments, our core contributions have been explicitly recognized by multiple reviewers:

- The framework is simple, well-designed, and easy to implement.

- The motivation for removing AE is clear, and the actual results meet expectations.

- The Anchor Token selection strategy is intuitive, explainable, and easy to implement.

- Comparisons of SAC with baseline methods are comprehensive, and SAC demonstrates superior performance across multiple QA tasks.

## II. Main Questions Raised by Reviewers

Despite the overall positive feedback, some reviewers have raised the following constructive concerns:

- Whether the SAC method can demonstrate broader applicability across additional tasks.

- Whether the SAC method can be scaled to larger models.

- Comparisons with more recent baselines (DAST[5], Activation Beacon[6]).

- Further explanations and analysis on why including AE hurts downstream task performance.

## III. Supplementary Experiments and Progress in the Rebuttal

To address the above concerns, we conducted the following additional experiments (including but not limited to):

- Extensive experiments on long-context QA and summarization tasks to validate the method's task generalization.

- Systematic scalability experiments on 1B / 3B / 8B models.

- Comparisons with the latest baselines (DAST[5], Activation Beacon[6]), where SAC consistently outperforms them.

- Verification of the effects of AE through intuitive analysis, experimental results, and visualization.

Through these additional materials, we have tried to address the reviewers’ core concerns (for detailed experimental results, please refer to the Supplementary Results).

Before the anomaly occurred on OpenReview:

- Reviewer wACd increased their score to 8 after reviewing our supplementary explanations and new experiments;

- Although the other reviewers did not provide further replies, We have provided additional supporting experiments and explanations in response to their questions.


We sincerely thank you again for taking the time to handle this submission under these special circumstances. Should you require any further experiments, analyses, or technical clarifications, we would be more than happy to provide them immediately.


[1] Tao Ge, et al. In-context Autoencoder for Context Compression in a Large Language Model. ICLR, 2024.

[2] Zongqian Li, et al. 500xCompressor: 500xCompressor: Generalized Prompt Compression for Large Language Models. ACL, 2025.

[3] Runsong Zhao, et al. Position IDs Matter: An Enhanced Position Layout for Efficient Context Compression in Large Language Models. EMNLP Findings, 2025.

[4] Xiangfeng Wang, et al. In-Context Former: Lightning-fast Compressing Context for Large Language Model. ACL Findings, 2024.

[5] Shaoshen Chen, et al. DAST: Context-Aware Compression in LLMs via Dynamic Allocation of Soft Tokens. ACL Findings, 2025.

[6] Zhang, Peitian, et al. Long Context Compression with Activation Beacon. ICLR, 2025.

---

### Meta-Review · Area_Chair_PtUK · 2026-01-07

**Summary:**

This paper proposes "Semantic-Anchor Compression" (SAC), a novel context compression framework that eliminates the autoencoding (AE) objective commonly used in prior work. The authors posit that AE tasks are misaligned with downstream generation capabilities and instead propose a method that selects "anchor tokens" from the original context, aggregating information via bidirectional attention optimized solely for the target task.

The initial reviews were mixed (scores of 6, 6, 4, 2), with reviewers raising valid concerns regarding the method's scalability beyond small models (1B parameters), potential overfitting to QA tasks due to the removal of the reconstruction objective, and the absence of recent baselines.

During the rebuttal, the authors provided a comprehensive set of additional experiments that addressed the primary concerns. Specifically, they scaled the method to Llama-3.2-3B and Llama-3.1-8B, expanded the evaluation to long-context summarization tasks (QMSum, GovReport), and implemented comparisons against suggested baselines (DAST, Activation Beacon). The new results demonstrate that SAC maintains performance advantages across these settings. Reviewer wACd explicitly raised their score to 8 following these updates. While Reviewer 4eQ5 maintained a lower score, their review appears to stem from a misunderstanding of the distinction between training-free pruning and training-based compression. Given the robust empirical evidence and the successful rebuttal, I recommend acceptance.

**Reviewer Concerns:**

Addressed Concerns:

Scalability: Reviewers rbBK and wACd questioned the reliance on a 1B parameter model. The authors conducted new experiments on 3B and 8B models, showing consistent improvements over the EPL baseline as model size increases.

Generalization to Non-QA Tasks: Reviewer NhSy raised concerns that removing the AE objective might degrade performance on tasks requiring holistic context understanding, such as summarization. The authors added experiments on QMSum and GovReport, where SAC remained competitive and outperformed baselines.

Missing Baselines: Reviewers requested comparisons to DAST and Activation Beacon. The authors implemented these methods and demonstrated that SAC achieves higher performance on both in-domain and out-of-domain benchmarks.

Validation of "AE-Free" Hypothesis: The authors provided ablation studies comparing SAC against a variant trained with AE loss (SAC+AE), showing that the inclusion of AE loss degrades downstream performance, supporting their central hypothesis.

Outstanding Concerns:

Inference Latency: While the authors provided a theoretical FLOPs analysis indicating SAC is efficient due to reduced sequence lengths, a direct empirical measurement of wall-clock latency including the bidirectional attention overhead remains a minor missing data point.

**Reviewer Scores:**

Reviewer wACd: This reviewer was highly responsive to the rebuttal. They raised their score after the authors provided the requested trade-off curves, initialization details, and additional baselines.

Reviewer NhSy: Originally concerned about overfitting to QA tasks. The inclusion of summarization results (GovReport, QMSum) addressed this concern directly, justifying a positive score.

Reviewer rbBK: The initial score was based on clarity issues and limited model scale (1B). The authors clarified the method and provided results on 8B models and 24K context lengths, effectively resolving the grounds for the lower score.

Reviewer 4eQ5: This reviewer's critique centered on confusion between "context-agnostic" initialization and context-dependent attention, and conflated the method with training-free KV pruning approaches. The authors provided a clear clarification, suggesting the low score is based on a misunderstanding of the problem setting.

---

### Decision · Program_Chairs · 2026-01-26

Accept (Poster)